# Six New Species of *Agaricus* (Agaricaceae, Agaricales) from Northeast China

**DOI:** 10.3390/jof10010059

**Published:** 2024-01-11

**Authors:** Shi-En Wang, Tolgor Bau

**Affiliations:** Key Laboratory of Edible Fungal Resources and Utilization (North), Ministry of Agriculture and Rural Affairs, Jilin Agricultural University, Changchun 130118, China; yinxin2022@126.com

**Keywords:** morphology, multi-gene phylogeny, taxonomy, new species

## Abstract

*Agaricus* belongs to Agaricaceae and is one of the most well-known macrofungi, with many edible species. More than 300 *Agaricus* specimens were collected during a three-year macrofungal resource field investigation in Northeast China. Based on morphological observations and multi-gene (ITS + nrLSU + *tef1-a*) phylogenetic analyses, six new *Agaricus* species, i.e., *Agaricus aurantipileatus*, *A. daqinggouensis*, *A. floccularis*, *A. griseopileatus*, *A. sinoagrocyboides*, and *A. velutinosus*, were discovered. These new species belong to four sections within different subgenera: *A*. (subg. *Agaricus*) sect. *Agaricus*, *A*. (subg. *Flavoagaricus*) sect. *Arvenses*, *A*. (subg. *Minores*) sect. *Minores*, and *A*. (subg. *Pseudochitonia*) sect. *Xanthodermatei*. Morphological descriptions, line illustrations, and basidiomata photographs of these new species are provided, and their differences from similar species are discussed.

## 1. Introduction

*Agaricus* L. is a large genus of the family Agaricaceae, with *Agaricus campestris* L. as the type; it can be distinguished from other genera by its unique characteristics, which include small-sized to large-sized fleshy basidiomata, free lamellae that are white or pink when young but at maturity become brown to dark brown, presence of an annulus on the stipe, brown basidiospores, brown spore prints, pileipellis a cutis of cylindrical hyphae, and absence of capitate cheilocystidia [1,2,3,4,5]. The species of *Agaricus* are saprophytic, distributed on all continents except Antarctica, and commonly found in forests, pastures, or grasslands [6,7].

*Agaricus* was first described by Linnaeus in 1753; Linnaeus collectively referred to all the large agaric mushrooms found at the time as ‘*Agaricus*’ [5]. However, the currently defined genus *Agaricus* was formally established by Karsten (1879) [8]. The taxonomic study of *Agaricus* has a long history, and different taxonomists hold different taxonomy views on the taxonomy system of *Agaricus*. Before the application of molecular systematics, the taxonomy system proposed by Parra (2008) was widely accepted based on morphology [9]. Parra (2008) summarized the previous studies and divided *Agaricus* into three subgenera: *A*. subg. *Agaricus*, *A*. subg. *Conioagaricus* and *A*. subg. *Lanagaricus* and eight sections, viz., *A*. sect. *Agaricus*, *A*. sect. *Arvenses*, *A*. sect. *Bivelares*, *A*. sect. *Chitonioides*, *A*. sect. *Minores*, *A*. sect. *Sanguinolenti*, *A*. sect. *Spissicaules*, and *A*. sect. *Xanthodermatei*, based on its macroscopic and microscopic characteristics [1].

In recent years, with the continuous development of biotechnology and scientific means, the taxonomy system of *Agaricus* has also tended to be stable. Based on the molecular identification of *Agaricus* species, Zhao et al. (2011) added sequences of specimens from tropical and temperate regions, and molecular phylogenetic results verified the eight known sections proposed by Parra (2008) and found that there were 11 independent clades [7]. Additionally, Zhao et al. (2016) proposed a new taxonomic system that classified *Agaricus* into five subgenera and 20 sections based on divergence times [2]. In subsequent studies, new subgenera and sections were established based on the new taxonomic system;presently, *Agaricus* comprises six subgenera and 27 sections [6,10,11,12,13,14].

There is a seventh subgenus, *A*. subg. *Conioagaricus*, containing three sections: *A*. sect. *intermedii*, *A*. sect. *Pulverotecti*, and *A*. sect. *Striati* [15]. However, due to the members of *A*. subg. *Coniagaricus* have an epithelium pileipellis, which is completely different from species of the other subgenera, *A*. subg. *Coniagaricus* is suggested to be relocated to other genera [2,16]. Unfortunately, in the absence of available molecular data, *A*. subg. *Conioagaricus* has not been studied molecularly within the framework of the new taxonomic system proposed by Zhao et al. (2016) [2]. *A*. subg. *Lanagaricus* [15] has been proved a heterotypic synonym of *A*. subg. *Pseudochitonia* [2].

This study involves four sections of four subgenera, viz., *A*. sect. *Agaricus* in *A*. subg. *Agaricus*, *A*. sect. *Arvenses* in *A*. subg. *Flavoagaricus*, *A*. sect. *Minores* in *A*. subg. *Minores* and *A*. sect. *Xanthodermatei* in *A*. subg. *Pseudochitonia*, in the system by Zhao et al. (2016).

*Agaricus* species are abundant; so far, more than 500 species have been recorded [1,2,3,7,17,18,19,20]. At present, over 116 Agaricus species supported by molecular phylogenetic and morphological studies have been reported in China [9,21,22,23,24,25,26,27]. However, the richness of *Agaricus* species diversity in some regions of China still needs to be studied and explored, such as Northeast China. Northeast China is mainly a north temperate zone, has a temperate monsoon climate with four distinct seasons [28] and abundant plant resources [29], which provides the basis for fungal diversity.

From 2021 to 2023, we collected more than 300 *Agaricus* specimens in Northeast China. Among those, after morphological study and phylogenetic analyses, 34 species were identified, including six new species. In this study, we describe these new species of *Agaricus* to better understand the species diversity of *Agaricus* in this region.

## 2. Material and Methods

### 2.1. Specimens and Morphological Observations

The examined specimens were mainly collected from Northeast China, including the Inner Mongolia Autonomous Region, Jilin Province, and Heilongjiang Province, and some were studied from the Herbarium of Mycology of Jilin Agricultural University (HMJAU). The voucher specimens are deposited in the Herbarium of Mycology of Jilin Agricultural University. The macromorphological characteristics are based on field records and photographs of fresh basidiomata. The color description of the fresh basidiomata is referenced by Kornerup and Wanscher (1978) [30]. Microscopic features were observed based on dry specimens. The corresponding structures were taken and prepared freehand, floated in 5% KOH solution or sterile water, stained with 1% Congo red solution if necessary, and observed through a light microscope (Olympus BX53, Olympus, Tokyo, Japan). The microscopic characteristics of each structure were based on at least 20 measurements. The symbol ‘(a) b–c (d)’ is used to describe the size of basidiospores, where the ‘b–c’ range represents 90% of the measured values, while the ‘a’ and ‘d’ are extreme values. ‘[Xav = e × f]’ indicates the average size of basidiospores. ‘Q’ refers to the ratio of length to width of a single basidiospore from the side view, and ‘Qav’ refers to the average value of ‘Q’ of all specimens. Other microstructural measurements include the range between the extreme length and width measurements.

### 2.2. DNA Extraction, PCR Amplification, and Sequencing

Genomic DNA was extracted using the NuClean Plant Genomic DNA kit (CWBIO, Beijing, China) in strict accordance with the instruction manual. The primer pairs ITS1F/ITS4 [31], LR0R/LR5 [32], and EF1-983F/EF1-1567R [33] were used to amplify the sequences of three DNA regions, ITS, nrLSU, and *tef1-a*, respectively. The polymerase chain reaction (PCR) procedure was based on the protocol described by Mou and Bau (2021) [34]. The PCR products were detected using 1% agarose gel electrophoresis, and the qualified products were sent to Bioengineering (Shanghai) Co., Ltd., Shanghai, China, for sequencing with the same primers.

### 2.3. Sequence Alignment and Phylogenetic Analyses

The chromatograms were checked in BioEdit v.7.1.3.0 [35] to ensure that each sequence had good quality. Then, a BLAST search was carried out in the National Center of Biotechnology Information (NCBI) database (https://www.ncbi.nlm.nih.gov/ (accessed on 19–24 October 2023)) to confirm the sequencing results. Finally, the sequences were submitted to GenBank (Table 1 in bold). Based on the BLAST search results, sequences corresponding to the subgenera of the studied species were downloaded for phylogenetic analyses. Subsequently, the multi-gene phylogenetic trees of these subgenera were constructed separately. Particularly, species (Table 1) falling within the clades of the species described in this study were selected and integrated to construct new multi-gene phylogenetic trees. *Heinemannomyces* sp. ZRL185 was used as an outgroup [2,13].

Sequences alignment was performed using the online MAFFT tool [36] (https://mafft.cbrc.jp/alignment/software/, accessed on 8–17 October 2023) to align sequences independently for each region, then manually adjusted in BioEdit v.7.1.3.0. The ITS, nrLSU, and *tef1-a* were assembled in Phylosuite v1.2.2 [37]. The multi-locus dataset (ITS + nrLSU + *tef1-a*) of *Agaricus* had an aligned length of 2107 (ITS subset: 1–730 bp; nrLSU subset: 731–1594 bp; *tef1-a* subset: 1595–2107 bp) total characters including gaps. The alignment was submitted to Figshare (https://doi.org/10.6084/m9.figshare.24631926.v3, accessed on 7 January 2024).

Maximum likelihood (ML) phylogenies were inferred using IQ-TREE [38] under the TIM2 + I + G4 + F model for 5000 ultrafast bootstraps [39], as well as the Shimodaira–Hasegawa–like approximate likelihood-ratio test. ModelFinder [40] was used to select the best-fit model of ML phylogenies using the BIC criterion. Bayesian Inference (BI) phylogenies were inferred using MrBayes 3.2.6 [41] under the partition model (2 parallel runs, 743,400 generations), in which the initial 25% of sampled data were discarded as burn-in. ModelFinder was again used to select the best-fit partition model (Edge-linked) using the BIC criterion, and the best-fit model according to BIC was HKY + F + G4 for ITS, HKY + F + I for nrLSU and K2P + I for *tef1-a*. The final trees were visualized using iTOL [42] and edited using Adobe Illustrator 2021 (Adobe, San Jose, CA, USA).

**Table 1 jof-10-00059-t001:** Sequences used in the phylogenetic analysis. ‘T’ refers to the type specimen. Bold refers to the sequences produced from this study. Green font refers to the new species. ‘–’ means no relevant genetic information.

Taxon	Voucher ID	Origin	ITS	nrLSU	*tef1-* *α*	References
*A.* sect*. Agaricus*						
*Agaricus* *albovariabilis*	TBGT18462	India	ON555770	–	–	[43]
*A. albovariabilis*	TBGT18487 T	India	ON555779	–	–	[43]
*A. argyropotamicus*	RWK2017	The USA	KJ877748	–	–	[3]
*A. argyropotamicus*	F2047	France	JF727849	–	–	[24]
*A. flavicentrus*	MFLU12-0146 T	Thailand	NR151752	–	–	[24]
*A. flavicentrus*	MFLU12-0149	Thailand	KR025856	–	–	[24]
*A. inilleasper*	H4452 T	Australia	NR119947	–	–	[24]
** * A. sinoagrocyboides * **	** HMJAU 67738 T **	** China **	** OR690292 **	** OR690373 **	** OR711549 **	** In this study **
** * A. sinoagrocyboides * **	** HMJAU 46513 **	** China **	** OR690290 **	** OR690371 **	** OR711547 **	** In this study **
** * A. sinoagrocyboides * **	** HMJAU 46514 **	** China **	** OR690291 **	** OR690372 **	** OR711548 **	** In this study **
*A.* sp.	F2272	France	JF727850	–	–	[7]
*A.* sect*. Arvenses*						
*A. abruptibulbus*	LAPAG524	Czech Republic	KJ548132	–	–	[17]
*A. abruptibulbus*	ZRL20161250	China	MK617909	MK617821	MK614417	[25]
** *A. abruptibulbus* **	**HMJAU 67833**	**China**	**OR690278**	–	–	**In this study**
** *A. abruptibulbus* **	**HMJAU 67834**	**China**	**OR690279**	–	–	**In this study**
*A. arvensis*	RWK2287	the USA	KJ847460	–	KX198047	[3]
*A. arvensis*	LAPAG450 T	Spain	KT951328	KP739801	KT951619	[2]
** * A. aurantipileatus * **	** HMJAU 67746 **	** China **	** OR690304 **	** OR690379 **	** OR711537 **	** In this study **
** * A. aurantipileatus * **	** HMJAU 67747 T **	** China **	** OR690305 **	** OR690380 **	** OR711538 **	** In this study **
*A. fissuratus*	ZRL20170745	China	MK617935	MK617847	MK614441	[25]
*A. fissuratus*	RWK2285	the USA	KJ859085	–	–	[3]
** * A. floccularis * **	** HMJAU 67744 T **	** China **	** OR690297 **	** OR690377 **	** OR711535 **	** In this study **
** * A. floccularis * **	** HMJAU 67745 **	** China **	** OR690298 **	** OR690378 **	** OR711536 **	** In this study **
*A. greuteri*	PALGR57140 T	Italy	KF114473	–	–	[17]
*A. greuteri*	LAPAG399	Spain	MK617859	–	–	[25]
** * A. griseopileatus * **	** HMJAU 67838 **	** China **	** OR690299 **	** OR690381 **	** OR711539 **	** In this study **
** * A. griseopileatus * **	** HMJAU 67841 **	** China **	** OR690300 **	** OR690382 **	** OR711540 **	** In this study **
** * A. griseopileatus * **	** HMJAU 67848 T **	** China **	** OR690301 **	** OR690383 **	** OR711541 **	** In this study **
** * A. griseopileatus * **	** HMJAU 67852 **	** China **	** OR690302 **	** OR690384 **	** OR711542 **	** In this study **
** * A. griseopileatus * **	** HMJAU 67856 **	** China **	** OR690303 **	** OR690385 **	** OR711543 **	** In this study **
*A. guizhouensis*	HKAS 81081 T	China	KJ755658	–	*–*	[44]
*A. guizhouensis*	ZRL20160420	China	MK617906	MK617818	MK614414	[25]
** *A. guizhouensis* **	**HMJAU 67830**	**China**	**OR690276**	–	–	**In this study**
** *A. guizhouensis* **	**HMJAU 67831**	**China**	**OR690277**	–	–	**In this study**
*A. luteopileus*	ZRL2012604 T	China	KT951375	KT951515	KT951620	[2]
*A. luteopileus*	ZRL20120589	China	MK617868	MK617780	MK614377	[25]
*A. macrolepis*	LAPAG409	Spain	KJ548128	–	–	[17]
*A. megacarpus*	QL20170178 T	China	MK617861	MK617774	MK614371	[25]
*A. megacarpus*	ZRL20152608	China	MK617901	MK617813	MK614409	[25]
*A. reducibulbus*	SFSUF020926 T	the USA	NR_144994	–	–	[3]
*A. reducibulbus*	RWK2105	the USA	KJ859130	–	–	[3]
*A.* sp.	ZRL20170661	China	MK617927	MK617839	MK614434	[25]
*A.* sp.	ZRL2012630	China	KT951379	KT951495	KT951621	[2]
*A.* sp.	ZRL20162213	China	MK617916	MK617828	MK614423	[25]
*A.* sp.	SHY2011073117	China	KT951407	KT951459	KT951622	[2]
***A.* sp.**	**HMJAU 67829**	**China**	**OR690281**	–	–	**In this study**
** *A.* ** **sp.**	**HMJAU 67828**	**China**	**OR690282**	–	–	**In this study**
*A. subumbonatus*	ZRL2012030 T	China	KT951364	KT951455	KT951628	[2]
*A. subumbonatus*	ZRL20130529	China	MK617872	MK617784	MK614380	[25]
*A.* sect*. Minores*						
*A. fulvoaurantiacus*	MFLU16-0980 T	Thailand	KU975107	KX084002	KX198069	[6]
*A. fulvoaurantiacus*	MFLU16-0974	Thailand	KU975106	KX084001	KX198043	[6]
*A. huijsmanii*	LAPAG639	Spain	KF447889	KT951444	KT951571	[17]
*A. huijsmanii*	HMJAU 67712	China	OR159474	–	–	[27]
*A. huijsmanii*	HMJAU 67713	China	OR159473	–	–	[27]
*A. luteofibrillosus*	MFLU16-0981	Thailand	KU975108	KX084003	KX198041	[6]
*A. luteofibrillosus*	ZRL2110	Thailand	KU975109	KX084004	–	[6]
*A. rufifibrillosus*	ZRL20151536 T	China	KX684878	KX684893	KX684915	[45]
*A.* sp.	ZRL2010079	China	KX657046	KX656951	KX684950	[45]
*A.* sp.	HMJAU 67724	China	OR159498	OR144156	OR166452	[27]
*A.* sp.	HMJAU 67725	China	OR159497	OR144157	OR166451	[27]
** * A. velutinosus * **	**HMJAU 67768 T**	**China**	**OR690296**	**OR690376**	**OR711546**	**In this study**
** * A. velutinosus * **	**HMJAU 67763**	**China**	**OR690293**	**OR690374**	**OR711544**	**In this study**
** * A. velutinosus * **	**HMJAU 67764**	**China**	**OR690294**	**OR690375**	**OR711545**	**In this study**
** * A. velutinosus * **	**HMJAU 67767**	**China**	**OR690295**	**–**	**–**	**In this study**
*A*. sect. *Xanthodermatei*						
*A. atroumbonatus*	MM1637 T	Pakistan	MH997905	MK100290	MK169412	[46]
** *A. atroumbonatus* **	**HMJAU 67754**	**China**	**OR690273**	**OR690365**	**OR711550**	**In this study**
*A. berryessae*	RWK2106 T	The USA	KJ609482	–	–	[3]
*A. bisporiticus*	LD2012111 T	Thailand	KJ575611	KT951507	KT951650	[12]
*A. bisporiticus*	MCR25	Pakistan	KJ575608	–	–	[12]
*A. daliensis*	SHY2011071706	China	KM657877	KR006615	KR006643	[47]
** * A. daqinggouensis * **	**HMJAU 67756 T**	**China**	**OR690287**	**OR690369**	**OR711554**	**In this study**
** * A. daqinggouensis * **	**HMJAU 67758**	**China**	**OR690288**	**–**	**–**	**In this study**
** * A. daqinggouensis * **	**HMJAU 67759**	**China**	**OR690289**	**OR690370**	**OR711555**	**In this study**
*A. deardorffensis*	ecv4230 T	The USA	KJ609494	–	–	[3]
*A. moelleroides*	LAPAG319 T	Portugal	JN204437	–	–	[17]
*A. pseudopratensis*	LAPAG259	Spain	DQ182527	MK123324	MK169403	[46]
** *A. pseudopratensis* **	**HMJAU 67815**	**China**	**OR690274**	**–**	**–**	**In this study**
** *A. pseudopratensis* **	**HMJAU 67816**	**China**	**OR690275**	**–**	**–**	**In this study**
***A.* sp.**	**HMJAU 67739 **	**China**	**OR690283**	**OR690366**	**OR711551**	**In this study**
***A.* sp.**	**HMJAU 67740**	**China**	**OR690284**	**OR690367**	**OR711552**	**In this study**
***A.* sp.**	**HMJAU 67741**	**China**	**OR690285**	**OR690368**	**OR711553**	**In this study**
***A.* sp.**	**HMJAU 42077**	**China**	**OR690286**	**–**	**–**	**In this study**
***A.* sp.**	**HMJAU 68021**	**China**	**OR690280**	**–**	**–**	**In this study**
*A. tephrolepidus*	JBSD123822 T	Dominican Republic	MF511117	–	–	[12]
** *A. tephrolepidus* **	**HMJAU 67812**	**China**	**OR690269**	**OR690362**	**OR711556**	**In this study**
** *A. tephrolepidus* **	**HMJAU 67813**	**China**	**OR690270**	**OR690363**	**OR711557**	**In this study**
** *A. tephrolepidus* **	**HMJAU 67814**	**China**	**OR690271**	**OR690364**	**–**	**In this study**
** *A. tephrolepidus* **	**HMJAU 68018**	**China**	**OR690272**	**–**	**–**	**In this study**
*A. tibetensis*	ZRL2012585 T	China	KM657895	KR006633	KR006658	[47]
*Heinemannomyces*						
*Heinemannomyces* sp.	ZRL185	Thailand	KT951346	KT951527	KT951657	[2]

### 2.4. Species-Specific ITS Markers

The position of the unique nucleotide (nt) base of a species’ ITS sequence in the section to which the species belongs is represented as follows: ‘xxxxxXxxxxx @ position’, where the capital letter ‘X’ represents the exclusive or informative character, ‘@ position’ represents the position of ‘X’, and ‘xxxxx’ represents the flanking characters. The positions of ITS markers are sequentially numbered starting from the 5′ end (ggaaggatcatta). The insertion or deletion (indel) in the ITS alignment was disregarded rather than numbered. It must be noted that the comparisons are made with the currently available sequences of all species in the section and may need to be reassessed when the number of species changes.

## 3. Results

### 3.1. Molecular Phylogeny

The dataset used for the phylogenetic analysis consisted of 90 specimens (see Table 1), and 84 sequences were generated for this study, including 37 ITS sequences, 24 nrLSU sequences, and 23 *tef1-a* sequences. BI and ML analysis resulted in a very similar topology, so the ML tree is provided in this study (Figure 1). Bootstrap support (BS) values ≥ 50% and Bayesian posterior probability (PP) values ≥ 0.70 are indicated on branches (BS/PP).

The phylogenetic tree presents four main clades, corresponding to four sections of different subgenera. Six new species are distributed in these four sections as follows: *Agaricus daqinggouensis* and *A. moelleroides* LAPAG319 formed a sister clade with an appreciable support value (BS/PP = 84/0.80) in *A*. sect. *Xanthidermatei*. *Agaricus sinoagrocyboides* and *A. albovariabilis* TBGT18487/TBGT18462 formed a sister clade with a support value (BS/PP = 97/1) in *A*. sect. *Agaricus*. *Agaricus velutinosus* and *A. huijsmanii* LAPAG639/HMJAU67712/HMJAU67713 formed a sister clade with a support value (BS/PP = 100/1) in *A*. sect. *Minores*. *Agaricus floccularis* and *A. fissuratus* ZRL20170745/RWK 2285 formed a sister clade with a support value (BS/PP = 100/1); *A. aurantipileatus* and *A.* sp. ZRL2012630 formed a sister clade with a support value (BS/PP = 100/1), and *A. griseopileatus* also formed a unique clade with a high support value (BS/PP = 90/0.97).

### 3.2. Taxonomy

***Agaricus sinoagrocyboides*** T. Bau & S.E. Wang, **sp. nov.** (Figure 2a,b and Figure 3)

MycoBank: MB 851075

Etymology: ‘*sinoagrocyboides*’ refers to the macroscopic characteristics of this Chinese species, which are similar to those of *Agrocybe* species.

Holotype: China, Anhui Province: Feidong county, Hefei City, 117°19′–117°52′ E, 31°34′–32°16′ N, alt. 28 m, 31 July 2022, Tolgor Bau and Hong Cheng, C22073101 (HMJAU 67738).

Diagnosis: *A. sinoagrocyboides* is characterized by a small to medium-sized yellowish white (3A2) basidiomata, pileus with white (2A1) fibrils and cheilocystidia absent.

**Pileus** 2.3–5.8 cm in diameter, convex to the plane, obtusely umbonate at the center, surface dry, white (2A1), yellowish white (3A2), yellowish pale (2A3), with white (2A1) fibrils (especially at the margin), sometimes wrinkled, with an appendiculate margin by annulus remnants. **Lamellae** 0.2–0.6 cm broad, brownish grey (11F2), then chestnut (6F7) to brownish black (8F8), free, crowded, intercalated with numerous lamellulae. **Stipe** 3.5–5.2 cm long, 0.3–0.9 cm thick, nearly cylindrical with bulbous base, yellowish white (3A2), dull yellow (3B3) or straw yellow (3B4), hollow, nearly smooth, or below the annulus with white (2A1) fibrils. **Annulus** superior, simple, membranous, white (2A1), easy falling out. **Context** white (2A1), thin, no special odor.

**Basidiospores** (5.8) 6.0–7.4 (7.6) × (3.9) 4.2–5.1 (5.2) μm, [Xav = 6.9 × 4.6], Q = 1.36–1.66, Qav = 1.49, ellipsoid to elongate-ellipsoid, smooth, brown, thick-walled, guttulate. **Basidia** 18–26 × 6–8 μm, clavate, 4(2)-spored, sterigmata 2–3 µm long. **Cheilocystidia** absent. **Pleurocystidia** absent. **Pileipellis** a cutis of cylindrical hyphae, 5–10 μm wide, smooth, hyaline, slightly constricted at the septa.

Habit, habitat, and distribution: Solitary or scattered in grassland in summer. Currently, it is only known from the Anhui and Liaoning Provinces, China.

Species-specific ITS markers in *A*. sect. *Agaricus*: gtcttCggttg@113, gcctgTcgggg@219, and tggcgCgggga@608.

Additional specimens examined: China, Liaoning Province: Economic and Technological Development Zones, Dalian City, 2 July 2017, Tolgor Bau, HMJAU 46513, HMJAU 46514.

Notes: *A. sinoagrocyboides* belongs to *A*. sect. *Agaricus*. Before *A. sinoagrocyboides*, five species of this section have been found in China: *A. argenteus* Braendle ex Peck, *A. aristocratus* Gulden, *A. griseicephalus* Kerrigan, *A. jilinensis* R.L. Zhao &A.Q. Liu, and *A. zhangyensis* R.L. Zhao & A.Q. Liu [24]. *Agaricus argenteus*, *A. aristocratus,* and *A. zhangyensis* differ in having a white pileus; *A. griseicephalus* has a smooth, light brown pileus at maturity and smaller Qav (Qav = 1.3); and *A. jilinensis* in having a dark brown, with reddish–brown fibrille squamules pileus [24].

In the phylogenetic tree (Figure 1), *A. sinoagrocyboides* and *A.* sp. F2272, the sequence uploaded by Zhao et al. (2011) [7], formed a clade. *Agaricus albovariabilis* C.P. Arya & C.K. Pradeep, and *A. sinoagrocyboides* formed a sister clade with a high support value. However, *A. albovariabilis* possesses white basidiomata, becoming brownish at maturity, larger basidia (16–31 × 7.2–10.4 μm), and clavate to vesiculose clavate cheilocystidia [43].

*Agaricus agrocyboides* Heinem. and Gooss.-Font. was found in index Fungorum (http://indexfungorum.org/Names/Names.asp (accessed on 31 October 2023)). Although no molecular information on *A. agrocyboides* is available, it differs from *A. sinoagrocyboides* in having a brown pileus, slightly pink at the margin, with concolor scales, longer stipe (6–8 cm) with small ochre scales below the annulus, narrower basidiospores (5.2–6.6 × 3.2–3.8 μm), smaller basidia (15–18 × 5.5–7 μm) and abundant, pyriform cheilocystidia [15]. According to Zhao et al. [48], *A. agrocyboides* belongs to the tropical species of *A.* sect. *Arvenses*.

***Agaricus aurantipileatus*** T. Bau & S.E. Wang, **sp. nov.** (Figure 2c,d and Figure 4)

MycoBank: MB 851079

Etymology: ‘*aurantipileatus*’ refers to the orange (5A8) and brownish yellow (5C8) floccose squamules on the pileus of this species.

Holotype: China, Jilin Province: Zhuque Mountain, Jilin City, 126°40′ 52″ E, 43°46′ 46″ N, alt. 309.7 m, 19 July 2023, Tolgor Bau and Xia Wang, W23071914 (HMJAU 67747).

Diagnosis: *A. aurantipileatus* is characterized by its pileus with orange (5A8), brownish yellow (5C8), concentrically arranged floccose squamules, reddish brown (8E8), or slightly purplish pink (14A3), stipe with bulbous base, and context unchanging when cut, and lamellae pale red (10A3).

**Pileus** 4.1–9.0 cm in diameter, hemispherical, convex to plane, slightly depressed at the center, surface dry, white (5A1) or orange white (5A2), covered with orange (5A8) or orange white (5A2), covered with orange (5A8) or brownish yellow (5C8) floccose squamules, concentrically arranged, denser at the brownish yellow (5C8) or topaz (5C5) disc, margin appendiculate by annulus remnants. **Lamellae** 0.3–0.5 cm broad, pale red (10A3) when young, reddish brown (8E8) or slightly purplish pink (14A3) at mature, free, crowded, intercalated with numerous lamellulae. **Stipe** 9.0–9.5 cm long, 0.6–1.9 cm thick, nearly cylindrical with bulbous base, with short white (5A1) rhizomorphs, hollow, white (5A1), pale red (10A3) above the annulus sometimes, fibrillose below the annulus. **Annulus** superior, double, membranous and pendant, slightly filamentous membranous near the stipe, white (5A1), flocculent squamules on the lower surface, persistent. **Context** of the pileus up to 0.6 cm thick, white (5A1), when cut unchanging, no special odor.

**Basidiospores** (5.2) 5.3–6.6 (6.8) × (4.0) 4.3–5.0 (5.1) μm, [Xav = 5.9 × 4.6], Q = 1.17–1.36, Qav = 1.28, broadly-ellipsoid to ellipsoid, smooth, thick-walled, brown, guttulate. **Basidia** 16–22 × 6–8 μm, clavate, 4(2)-spored, sterigmata 2–4 µm long, smooth, hyaline. **Cheilocystidia** abundant, 11–25 × 7–14 μm, nearly globose, oblong or sphaeropedunculate, and sometimes catenulate. **Pleurocystidia** absent. **Pileipellis** a cutis of cylindrical hyphae, 4–8 μm wide, light brown, slightly constricted at the septa.

Habit, habitat, and distribution: Gregarious or solitary in broad-leaved forests or coniferous and broad-leaved mixed forests in summer. Currently, it is only known from Heilongjiang and Jilin Provinces, China.

Species-specific ITS markers in *A*. sect. *Arvenses*: atgggCtgtag@36, tcatcTtgtca@140.

Additional specimens examined: China, Heilongjiang Province: Ussuri River Forest Park, Hulin City, 31 July 2023, Liyang Zhu, Z23073101 (HMJAU 67746).

Notes: *A. aurantipileatus* belongs to *A*. sect. *Arvenses*. *Agaricus aurantipileatus* is similar to *A. guizhouensis* Y. Gui, Zuo Y. Liu & K.D. Hyde of this section. *Agaricus guizhouensis* differs in having a context turning yellow when cut, larger Qav (Qav = 1.68), and smaller cheilocystidia (3.6–16.6 × 3.2–14.3 μm) [44].

In the polygenic tree (Figure 1), *A. aurantipileatus*, *A. subumbonatus* R.L. Zhao & B. Cao and *A.* sp. ZRL2012630 formed a larger clade with a high support value. *Agaricus subumbonatus* possesses a smooth, fibrillose squamulose pileus with age, a cylindrical to slightly clavate stipe, and larger basidiospores (Q = 1.3–1.8) [25]. The sequence of *Agaricus* sp. ZRL2012630 was uploaded by Zhao et al. [2], but no formal description was provided.

Despite some differences in the molecular characters of the specimens HMJAU 67747 and HMJAU 67746, we did not find any obvious morphological differences between them. Therefore, both are here considered conspecific.

***Agaricus floccularis*** T. Bau & S.E. Wang, **sp. nov.** (Figure 2e,f and Figure 5)

MycoBank: MB 851078

Etymology: ‘*floccularis*’ refers to the pileus of this species with wheat yellow (4B5) to dull yellow (3B3) fibrillose floccose squamules.

Holotype: China, Inner Mongolia Autonomous Region: Horqin Left Middle Banner, Tongliao City, 123°16′53″ E, 43°47′42″ N, alt. 255.6 m, 21 August 2022, Tolgor Bau and Shi-En Wang, E2208251 (HMJAU 67744).

Diagnosis: Pileus with concentrically arranged, often triangular, fibrillose floccose, wheat yellow (4B5), dull yellow (3B3) squamules, cheilocystidia nearly globose or sphaeropedunculate, sometimes catenulate.

**Pileus** 5.0–17.5 cm in diameter, truncate conical to the plane, with, sometimes, slightly depressed center, white (4A1), yellowish white (4A2), wheat yellow (4B5), with concentrically arranged, often triangular, fibrillose floccose squamules, wheat yellow (4B5) or dull yellow (3B3), denser at the disc, the squamules sometimes fall off at maturity, becoming nearly smooth, margin brownish beige (6E3), appendiculate by annulus remnants. **Lamellae** 0.3–0.7 cm broad, reddish brown (8E8) to brownish black (8F8), free, crowded, intercalated with numerous lamellulae. **Stipe** 3.5–16.1 cm long, 0.7–3.7 cm thick, nearly cylindrical, or with a slightly bulbous base, sometimes with short, coarse white (4A1) rhizomorphs, white (4A1), yellowish white (4A2), straw yellow (3B4), hollow, dark yellow (4C8) on touching or bruising, smooth above the annulus, woolly-floccose squamules below the annulus.

**Annulus** superior, double, membranous, white (4A1), yellowish white (4A2), straw yellow (3B4), fibrillose on the lower surface, persistent. **Context** thick, white (4A1), with no special odor or bitter almond odor.

**Basidiospores** (6.2) 6.3–7.5 (7.6) × (4.5) 4.8–5.7 (5.9) μm, [Xav = 6.8 × 5.2], Q = 1.17–1.51, Qav = 1.32, broadly-ellipsoid to ellipsoid, smooth, thick-walled, brown, guttulate. **Basidia** 17–24 × 7–9 μm, clavate, 4(2)-spored, sterigmata 2–3 µm long, smooth, hyaline. **Cheilocystidia** abundant, 8–16 × 6–10 μm, nearly globose or sphaeropedunculate, sometimes catenulate. **Pleurocystidia** absent. **Pileipellis** a cutis of cylindrical hyphae, 5–10 μm wide, light brown, not constricted at the septa.

Habit, habitat, and distribution: Solitary in broad-leaved forests in summer and autumn. Currently, it is only known from the Inner Mongolia Autonomous Region and Jilin Province, China.

Species-specific ITS markers in *A*. sect. *Arvenses*: aaatcGctttc@211, gtctcAgtgag@647.

Additional specimens examined: China, Jilin Province: Jilin Agricultural University, Changchun City, 7 September 2022, Shi-En Wang, E2290705 (HMJAU 67745).

Notes: *A. floccularis* belongs to *A*. sect. *Arvenses*. Morphologically, *A. floccularis* is similar to *A. greuteri* L.A. Parra, Cappelli & Kerrigan. *Agaricus greuteri* differs in having a smooth or small squamules stipe provided with numerous white fine mycelial strands at the base and larger Qav (Qav = 1.48) [17].

In the phylogenetic tree (Figure 1), *A. fissuratus* F.H. Møller, *A. macrolepis* (Pilát & Pouzar) Boisselet & Courtec. and *A. floccularis* formed a larger clade with a well-supported value. *Agaricus fissuratus* differs in having an ochre to orange-red, often radially cracked pileus, larger basidiospores (Qav = 1.52), and larger (7–45 × 5–27 μm) spherical, pyriform, or broadly clavate cheilocystidia [17]. *Agaricus macrolepis* has a rusted ochre, often fissured pileus and larger (17–40 × 14–32 μm), pyriform, nearly spherical, or broadly clavate cheilocystidia [17].

It is worth mentioning that *A. fissuratus* is considered a synonym for *A. arvensis* Schaeff, in Index fungorum (http://indexfungorum.org/Names/Names.asp (accessed on 1 May 2023)). However, the study of Parra showed that *A. fissuratus* is an independent species [17]. Cao et al. [25] and Medel-Ortiz et al. [18] accepted this view. This study agrees with Parra’s view.

***Agaricus griseopileatus*** T. Bau & S.E. Wang, **sp. nov.** (Figure 2 g,h and Figure 6)

MycoBank: MB 851080

Etymology: ‘*griseopileatus*’ refers to the pastel grey (2C1) pileus of this species.

Holotype: China, Jilin Province: Jilin Agricultural University, Changchun City, 125°24′ 44″ E, 43°48′ 27″ N, alt. 279.7 m, 22 July 2023, Tolgor Bau and Shi-En Wang, E23072202 (HMJAU 67848).

Diagnosis: *A. griseopileatus* is characterized by its pileus with white (2A1), greyish white (2B1) fibrils when young, nearly smooth at maturity, lamellae first reddish grey (10B2), then reddish brown (8E8) to brownish black (8F8), stipe with fibrillose below the annulus (especially at the base), and context unchanging when cut.

**Pileus** 2.0–11.1 cm in diameter, hemispherical, convex to plane, white (2A1), pastel grey (2C1), yellowish pale (2A3) at the center, with white (2A1), greyish white (2B1) fibrils when young, nearly smooth at maturity, sometimes cracked, slightly longitudinally striated, and margin appendiculate by annulus remnants. **Lamellae** 0.3–0.9 cm broad, reddish grey (10B2) initially, reddish brown (8E8) to brownish black (8F8), free, crowded, intercalated with numerous lamellulae. **Stipe** 7.5–12.8 cm long, 0.7–2.4 cm thick, nearly cylindrical, or with bulbous base, sometimes with white (2A1) rhizomorphs, hollow, white (2A1), yellowish white (2A2), fibrillose below the annulus (especially at the base). **Annulus** superior, double, white (2A1), yellowish white (2A2) or brownish black (8F8), membranous, or filamentous towards its insertion to the stipe, with flocculent squamules on the lower surface, cogwheel-like at the margin, persistent. **Context** of the pileus up to 1.0 cm thick, white (2A1), unchanging when cut, and no special odor.

**Basidiospores** (5.7) 5.8–7.3 (7.4) × (4.1) 4.2–5.4 (5.6) μm, [Xav = 6.6 × 4.8], Q = 1.24–1.58, Qav = 1.39, broadly-ellipsoid to ellipsoid, smooth, thick-walled, brown, guttulate. **Basidia** 15–24 × 7–9 μm, clavate, 4(2)-spored, sterigmata 2–3 µm long, smooth, hyaline. **Cheilocystidia** abundant, 7–19 × 6–16 μm, nearly globose, oblong, sometimes catenulate or clustered. **Pleurocystidia** absent. **Pileipellis** a cutis of cylindrical hyphae, 5–8 μm wide, hyaline, slightly constricted or not at the septa.

Habit, habitat, and distribution: Gregarious or solitary in broad-leaved forest or coniferous and broad-leaved mixed forest in summer and autumn. Currently, it is only known from the Inner Mongolia Autonomous Region and Jilin Province, China.

Species-specific ITS markers in *A*. sect. *Arvenses*: tttctGaatgg@29, ttgagCtagga@121, and ctactCttgaa@666.

Additional material examined: China, Inner Mongolia Autonomous Region: Tuquan County, Hinggan League, 26 August 2023, Qianru Liu, R238133 (HMJAU 67861). China, Jilin Province: Jingyuetan National Forest Park, Changchun City, 26 September 2020, Qingqing Dong, 21826DQQ4 (HMJAU 67852), same location, 22 August 2022, Shi-En Wang, E2208183 (HMJAU 67853); Tongyu County, Baicheng City, 19 August 2021, Tolgor Bau and Liyang Zhu, Z2181934 (HMJAU 67859); Jilin Agricultural University, Changchun City, 3 July 2022, Shi-En Wang, E22070316 (HMJAU 67838), same location, 11–12 July 2022, Shi-En Wang, E22071108 (HMJAU 67836), E22071204 (HMJAU 67841); Border of Songhua River, Jilin City, 19 July 2023, Xia Wang and Shi-En Wang, E2307159 (HMJAU 67855), W23072009 (HMJAU 67856); Entrepreneurial Reservoir, Taonan City, 26 August 2023, Shi-En Wang, E2308356 (HMJAU 67857).

Notes: *A. griseopileatus* belongs to the *A*. sect. *Arvenses*. *Agaricus griseopileatus* is similar to *A. arvensis* Schaeff. of this section. *Agaricus arvensis* has white or grayish white lamellae initially, an odor of anise or bitter almonds when the pileus margin is bruised, and context, sometimes, becoming yellow on cutting [17]. Although the macroscopic characteristics of both exhibit minimal differences, discernible disparities exist in their molecular characters.

In the polygenic tree (Figure 1), *A. griseopileatus* formed an independent clade with a high support value. *Agaricus abruptibulbus* Peck is a closely related species of *A. griseopileatus*, but *A. abruptibulbus* possesses a mostly pale-yellow pileus, sometimes with yellow floccose squamules, larger basidiospores Q (Q = 1.29–1.81), and larger (16–34 × 14–26 μm), spherical, pyriform or clavate cheilocystidia [17].

***Agaricus velutinosus*** T. Bau & S.E. Wang, **sp. nov.** (Figure 2i,j and Figure 7)

MycoBank: MB 851077

Etymology: ‘*velutinosus*’ refers to the fibrillose to minutely floccose white (2A1) pileus and stipe surface of this species with velutinous appearance.

Holotype: China, Jilin Province: Qianjin Experimental Forestry Farm, Jiaohe City, 127°42′18″ E, 43°57′25″ N, alt. 460.2 m, 25 August 2023, Tolgor Bau and Hong Cheng, C2382515 (HMJAU 67768).

Diagnosis: *A. velutinosus* is characterized by small-sized white (2A1) basidiomata, a white (2A1) fibrillose minutely floccose, looking similar to velutinous, pileus and stipe surfaces, lamellae white (2A1), then pastel grey (2C1), finally brownish grey (8C2), cheilocystidia clavate, broadly clavate, or globose with a long peduncle.

**Pileus** 1.2–2.4 cm in diameter, convex to plane, obtusely umbonate at the center, white (2A1), yellowish white (2A2) at the center, with white (2A1) looking similar to velutinous, margin appendiculate by annulus remnants. **Lamellae** 0.1–0.2 cm broad, white (2A1) to pastel grey (2C1) or brownish grey (8C2), free, crowded, intercalated with numerous lamellulae. **Stipe** 1.3–3.9 cm long, 0.2–0.8 cm thick, nearly cylindrical with bulbous base, white (2A1), greyish white (2B1), yellowish white (2A2), hollow, with white (2A1) slender rhizomorphs, with white (2A1), greyish white (2B1) fibrillose to minutely floccose white squamules below the annulus. **Annulus** superior, simple, white (2A1), greyish white (2B1), membranous, easy to crack radially, easy falling out. **Context** thin, white (2A1), no special odor.

**Basidiospores** (4.7) 4.9–5.8 (5.9) × (2.9) 3.0–3.7 (3.8) μm, [Xav = 5.2 × 3.4], Q = 1.35–1.76, Qav = 1.52, ellipsoid to elongate-ellipsoid, smooth, brown, thick-walled, guttulate. **Basidia** 14–20 × 5–7 μm, clavate, 4(2) spored, sterigmata 1–3 µm long. **Cheilocystidia** abundant, clavate, broadly clavate, or globose with a long peduncle, 19–32 × 7–13 μm. **Pleurocystidia** absent. **Pileipellis** a cutis of cylindrical hyphae, 5–11 μm wide, slightly constricted at the septa.

Habit, habitat, and distribution: Solitary or gregarious in broad-leaved forest in summer. Currently, it is only known from Jilin Province, China.

Species-specific ITS markers in *A*. sect. *Minores*: ttagaTttcat@76, gcaatCtgctg@151, gatgtAgggac@162, ggtttAtatgc@282, gaaccGgtttg@567, gataaCtatct@595, and ctaatAgtctc@644.

Additional specimens examined: China, Jilin Province: Qianjin Experimental Forestry Farm, Jiaohe City, 25 August 2023, Tolgor Bau and Xia Wang, W23082521 (HMJAU 67769); Red Rock National Forest Park, Huadian City, 28 August 2023, Tolgor Bau and Xianyan Zhou, Y2382816 (HMJAU 67762), Y2382817 (HMJAU 67763), Y2382818 (HMJAU 67764), Y2382822 (HMJAU 67765), Y2382823 (HMJAU 67766), Y2382824 (HMJAU 67767).

Notes: *A. velutinosus* belongs to *A*. sect. *Minores*. *Agaricus velutinosus* is similar to *A. pseudopallens* M.Q. He & R.L. Zhao, but *A. pseudopallens* has larger basidiomata (pileus diameter 2.3–3.8 cm), single, smooth annulus, and cheilocystidia absent [45].

In the phylogenetic tree (Figure 1), *A. velutinosus* and *A. huijsmanii* Courtec. formed a sister clade with a high support value. However, the pileus of *A. huijsmanii* is larger (1.4–4.0 cm), and cheilocystidia spherical and sometimes catenulate [17].

***Agaricus daqinggouensis*** T. Bau & S.E. Wang, **sp. nov.** (Figure 2 k,l and Figure 8)

MycoBank: MB 851073

Etymology: ‘*daqinggouensis*’ refers to the type locality, Daqinggou National Nature Reserve, of this species.

Holotype: China, Inner Mongolia Autonomous Region: Daqinggou National Nature Reserve, Tongliao City, 122°10′23.1″ E, 42°47′35.6″ N, alt. 208 m, 19 August 2022, Tolgor Bau and Shi-En Wang, E2208287 (HMJAU 67756).

Diagnosis: Pileus white (2A1) to greyish white (2B1), lamellae pale red (9A3), then reddish brown (8E8) and finally brownish black (8F8), stipe with bulbous base, becoming yellow (3B8) at the base on cutting, and cheilocystidia nearly globose or pyriform, sometimes catenulate.

**Pileus** 2.5–12.9 cm in diameter, hemispherical, truncate conical to the plane, slightly depressed at the center, surface dry, white (2A1), greyish white (2B1) or pastel grey (2C1), darker at the disc, nearly smooth, or with greyish white (2B1) fine fibrils, margin appendiculate by annulus remnants. **Lamellae** 0.4–0.9 cm broad, pale red (9A3), then reddish brown (8E8), finally brownish black (8F8), free, crowded, intercalated with numerous lamellulae. **Stipe** 4.0–12.8 cm long, 0.6–2.4 cm thick, nearly cylindrical with bulbous base, white (2A1) to honey yellow (4D6), hollow, sometimes with white (2A1) rhizomorphs, nearly smooth, or fibrillose below the annulus (especially at the base). **Annulus** superior, double, with irregular floccose squamules on the lower surface, white (2A1), then reddish brown (8E8), membranous, pendant, and persistent. **Context** of the pileus up to 1.4 cm thick, white (2A1), greyish white (2B1), and becoming yellow (3B8) at the stipe base on cutting, with no special odor.

**Basidiospores** (4.8) 5.0–6.2 (6.3) × (3.2) 3.3–3.4 (3.5) μm, [Xav = 5.5 × 3.4], Q = 1.38–1.82, Qav = 1.63, ellipsoid to elongate-ellipsoid, smooth, brown, thick-walled, guttulate. **Basidia** 13–19 × 5–7 μm, clavate, 4(2)-spored, sterigmata 1–3 µm long. **Cheilocystidia** 7–20 × 7–14 μm, abundant, nearly globose, or pyriform, sometimes catenulate. **Pleurocystidia** absent. **Pileipellis** a cutis of cylindrical hyphae, 5–12 μm wide, hyaline, slightly constricted at the septa.

Habit, habitat, and distribution: Gregarious or scattered in broad-leaved forests, coniferous forests, or grasslands in summer. Currently, it is only known from Jilin Province and Inner Mongolia Autonomous Region, China.

Species-specific ITS markers in *A*. sect. *Xanthodermatei*: tggatTtgagg@160, accctTtaaac@249, and gttttCacatg@271.

Additional specimens examined: China, Inner Mongolia Autonomous Region: Daqinggou National Nature Reserve, Tongliao City, 19 August 2022, Tolgor Bau and Shi-En Wang, E2208285 (HMJAU 67755), E2208295 (HMJAU 67757), same location, 16 July 2023, Tolgor Bau and Qianru Liu, R23796 (HMJAU 67760). China, Jilin Province: Hongye Valley, Jiaohe City, 24 July 2022, Tolgor Bau and Shi-En Wang, E220723 (HMJAU 67758); Longtan Mountain, Jilin City, 20 June 2023, Shi-En Wang, E2307182 (HMJAU 67759).

Notes: *A. daqinggouensis* belongs to *A*. sect. *Xanthodermatei*. The superficial morphological characteristics of *A. daqinggouensis* and *A. arvensis* Schaeff. are similar. However, *A. arvensis* differs in having basidiospores usually broadly ellipsoid to ellipsoid, rarely subglobose, smaller Qav (Qav = 1.34) and larger Xav (Xav = 6.9 × 5.18), and odor of anise or bitter almonds when the pileus margin is bruised [17].

In the phylogenetic tree (Figure 1), *A. berryessae* Kerrigan, *A. deardorffensis* Kerrigan, *A. moelleroides* Guinb. & L.A. Parra and *A. tibetensis* J.L. Zhou & R.L. Zhao are closely related species to *A. daqinggouensis*. However, *A. moelleroides* has a pileus with dense grain-brown squamules and white to brown lamellae [17]. *Agaricus berryessae* has a cylindrical stipe often tapered toward the base, odor mildly phenolic (stronger when dried), and larger basidia (19–26 × 5.5–7.5 μm) [3]. *Agaricus deardorffensis* has dark brown squamules at the center of the pileus and narrower cheilocystidia (11–18.5 × 4.5–8 μm) [3]. *Agaricus tibetensis* has dark brown squamules at the center of the pileus, and cheilocystidia is absent [47].

## 4. Discussion

The macroscopic and microscopic characteristics of many *Agaricus* species overlap; therefore, molecular data and phylogenetic analysis are necessary to identify *Agaricus* species that have similar morphological features [3,7,17]. *Agaricus griseopileatus*, a novel species examined in this study, serves as an exemplary illustration. The morphological distinction between *A. griseopileatus* and *A. arvensis* Schaeff. poses challenges, whereas their differentiation in phylogenetic analysis is straightforward. The exclusive base on ITS sequences for phylogenetic analysis of *Agaricus* is not methodologically rigorous [25]; for example, *A*. sect. *Arvenses* and *A*. sect. *Xanthodermatei*. The more DNA regions, nrLSU + *tef1-a*, should be added to ensure the accuracy of species of some sections identification and the stability of phylogenetic relationships.

*Agaricus daqinggouensis* belongs to *A*. sect. *Xanthodermatei.* However, certain species within *A*. sect. *Xanthodermatei* and *A*. sect. *Hondenses* exhibit toxicity, such as *A. xanthodermus* Genev., which can induce gastrointestinal symptoms [49,50,51]. Despite a lack of research, *A. daqinggouensis* is unsuitable for consumption due to potential toxicity. Because some *Agaricus* species are so similar in morphological characteristics, it is difficult to identify *Agaricus* species in the wild. Refraining from consuming unknown wild *Agaricus* species is advised as a precautionary measure against mushroom poisoning. Furthermore, there is a need for further investigation into the toxic components, mechanisms of poisoning, detoxification processes, and other aspects related to the toxicity of *Agaricus* species.

*Agaricus floccularis*, *A. aurantipileatus,* and *A. griseopileatus* are members of *A*. sect. *Arvenses*. Currently, *A*. sect. *Arvenses* has not been reported to contain any toxic species, with several exhibiting nutritional or medicinal potential, such as cultivated species including *A. arvensis* Schaeff., *A. augustus* Fr., and *A. subrufescens* Peck [3,17,25]. We postulate that *A. floccularis*, *A. aurantipileatus,* and *A. griseopileatus* are also edible species that may have potential medicinal and edible values as new cultivatable species. Particularly noteworthy, *A*. *griseopileatus* was reported to be edible by some locals, which is a precious fungal resource worthy of further excavation and development.

This study encompassed six new species in China, thereby enriching the species diversity of *Agaricus* in Northeast China. In the phylogenetic tree (Figure 1), some species could not be determined. For instance, *Agaricus* sp. HMJAU 67828, collected from Jilin Province, consisted of immature basidiomata, with only one specimen obtained. The findings of this study indicate the potential existence of undiscovered species in Northeast China needs to be studied further.

## Figures and Tables

**Figure 1 jof-10-00059-f001:**
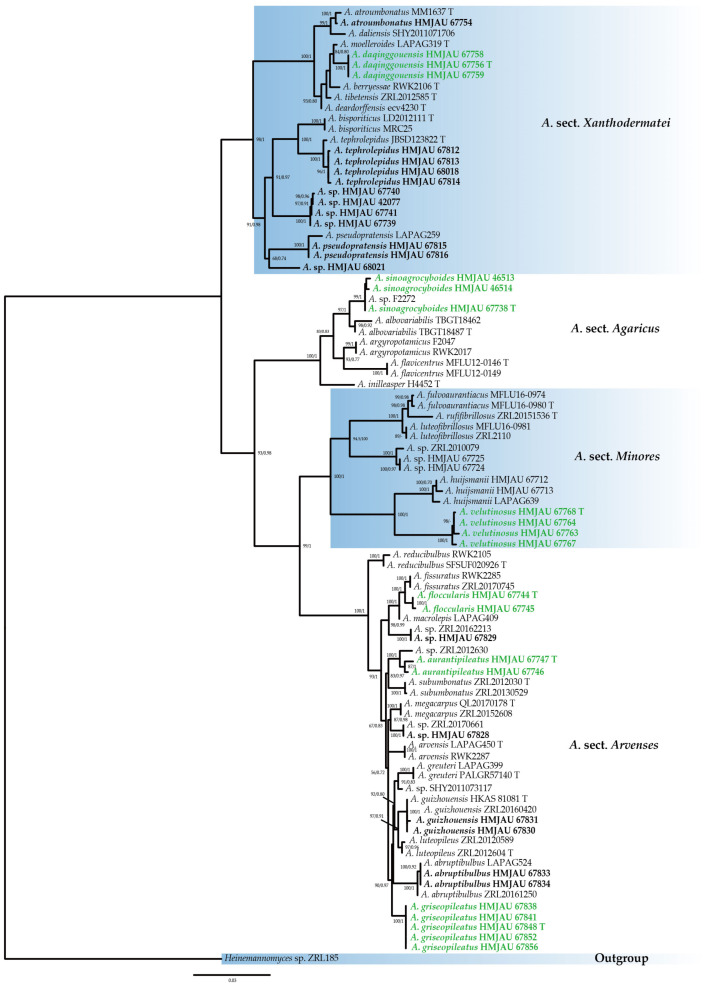
Multi-gene phylogenetic tree of *Agaricus* obtained from the maximum likelihood analysis (ML) based on ITS, nrLSU, and *tef1-a* sequence data. *Heinemannomyces* sp. ZRL185 was used as an outgroup. Bootstrap support (BS) values ≥ 50% and Bayesian posterior probability (PP) values ≥ 0.70 are indicated on branches (BS/PP). ‘T’ refers to the type specimen. Bold refers to the sequences produced from this study. Green font refers to the new species.

**Figure 2 jof-10-00059-f002:**
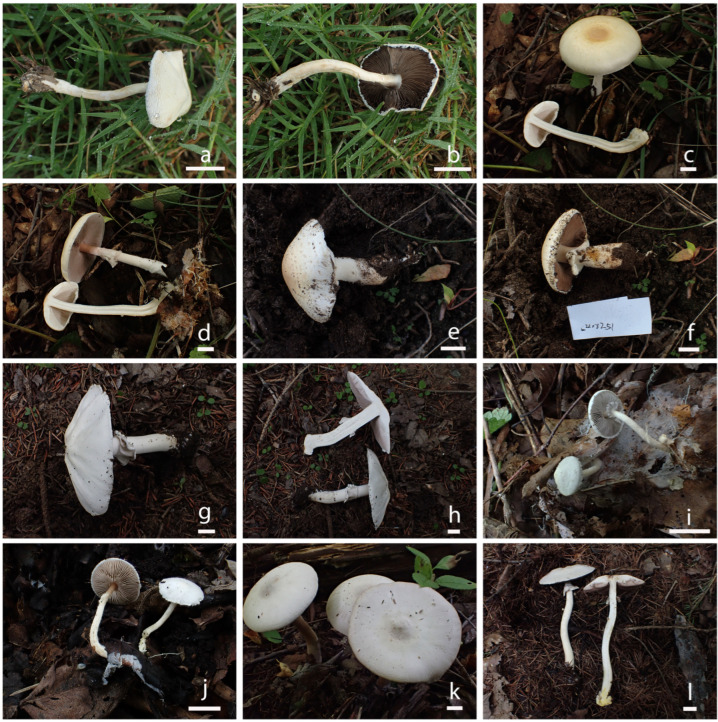
The photographs of fresh basidiomata of *Agaricus* species in this study. (**a**,**b**) *A. sinoagrocyboides* HMJAU 67738; (**c**,**d**) *A. aurantipileatus* HMJAU 67747; (**e**,**f**) *A. floccularis* HMJAU 67744; (**g**,**h**) *A. griseopileatus* HMJAU 67848; (**i**,**j**) *A. velutinosus* (**i**) HMJAU 67768, (**j**) HMJAU 67763; (**k**,**l**) *A. daqinggouensis* (**k**) HMJAU 67756, (**l**) HMJAU 67759. Bars: (**a**–**l**) = 1 cm.

**Figure 3 jof-10-00059-f003:**
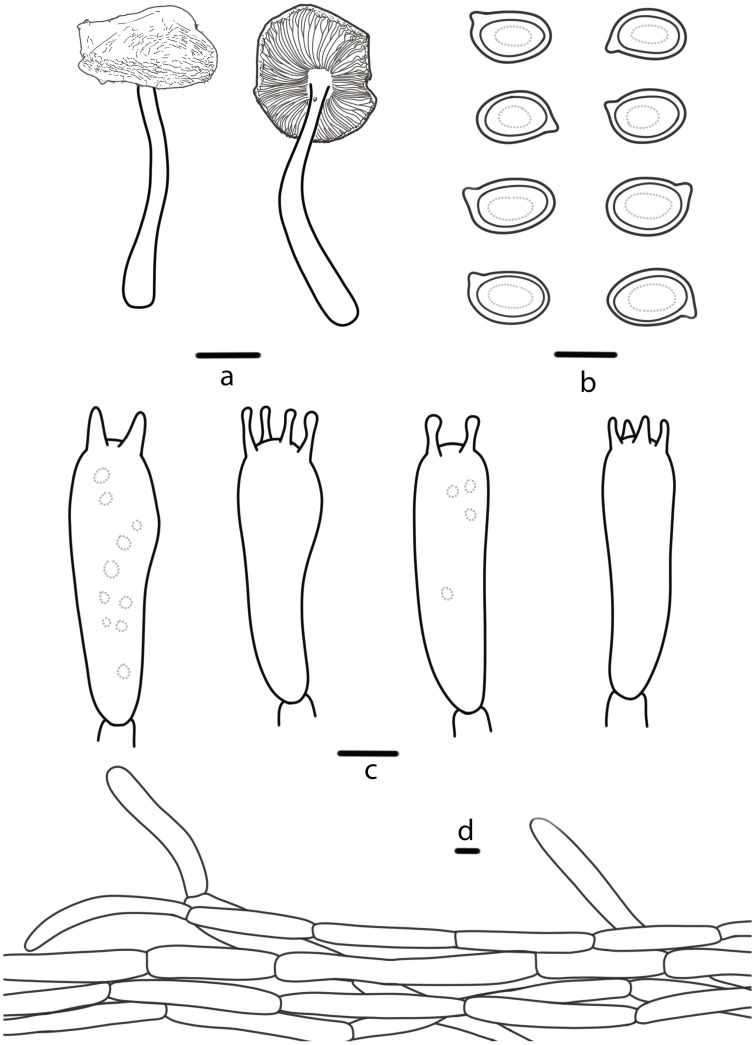
*Agaricus sinoagrocyboides* (HMJAU 46514, HMJAU 67738) (**a**) Basidiomata, (**b**) Basidiospores, (**c**) Basidia, (**d**) Pileipellis. Bars: (**a**) = 1 cm, (**b**–**d**) = 5 μm.

**Figure 4 jof-10-00059-f004:**
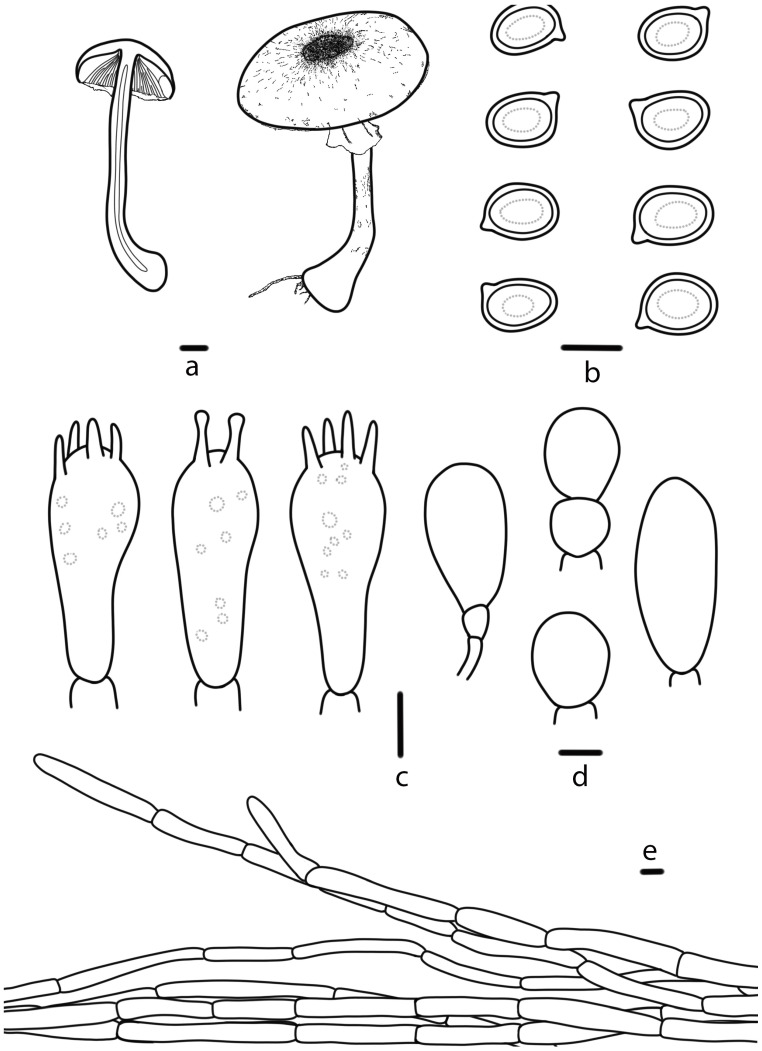
*Agaricus aurantipileatus* (HMJAU 67746, HMJAU 67747) (**a**) Basidiomata, (**b**) Basidiospores, (**c**) Basidia, (**d**) Cheilocystidia, (**e**) Pileipellis. Bars: (**a**) = 1 cm, (**b**–**e**) = 5 μm.

**Figure 5 jof-10-00059-f005:**
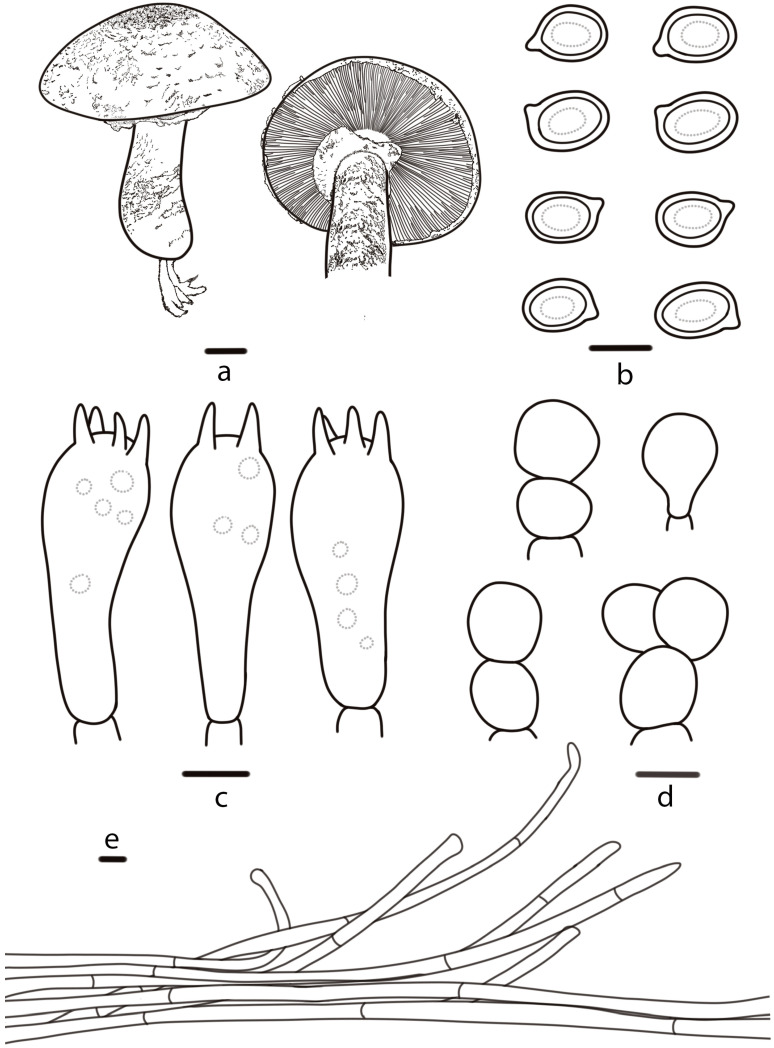
*Agaricus floccularis* (HMJAU 67744, HMJAU 67745) (**a**) Basidiomata, (**b**) Basidiospores, (**c**) Basidia, (**d**) Cheilocystidia, (**e**) Pileipellis. Bars: (**a**) = 1 cm, (**b**–**e**) = 5 μm.

**Figure 6 jof-10-00059-f006:**
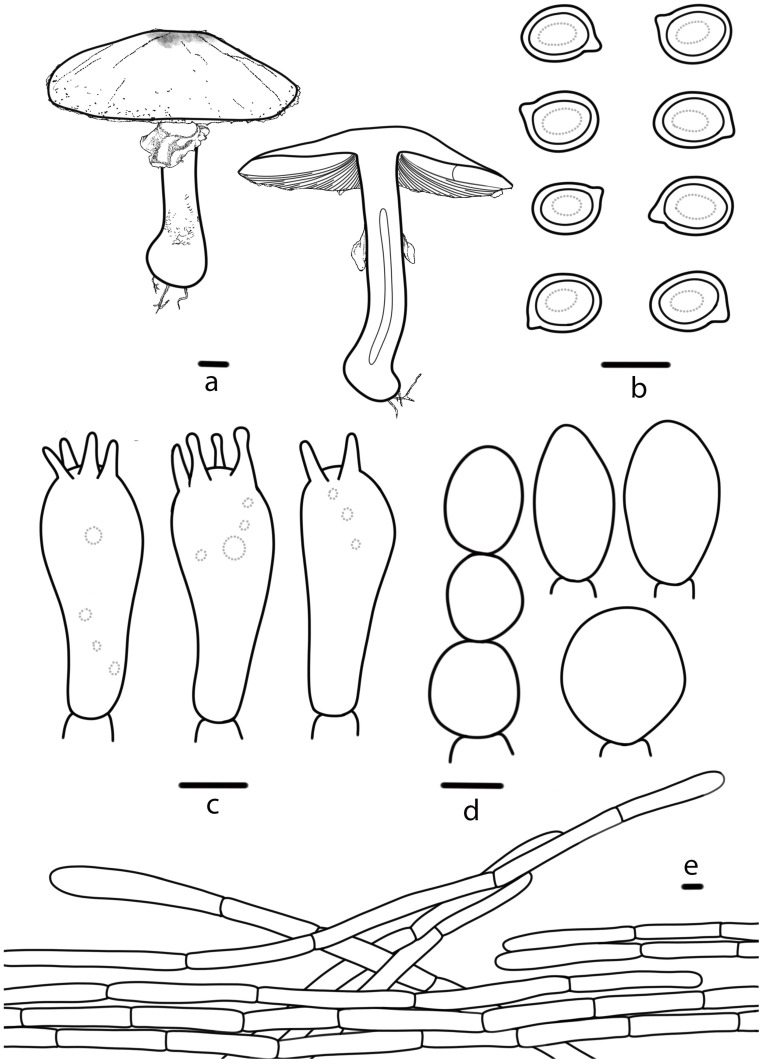
*Agaricus griseopileatus* (HMJAU 67838, HMJAU 67841, HMJAU 67848) (**a**) Basidiomata, (**b**) Basidiospores, (**c**) Basidia, (**d**) Cheilocystidia, (**e**) Pileipellis. Bars: (**a**) = 1 cm, (**b**–**e**) = 5 μm.

**Figure 7 jof-10-00059-f007:**
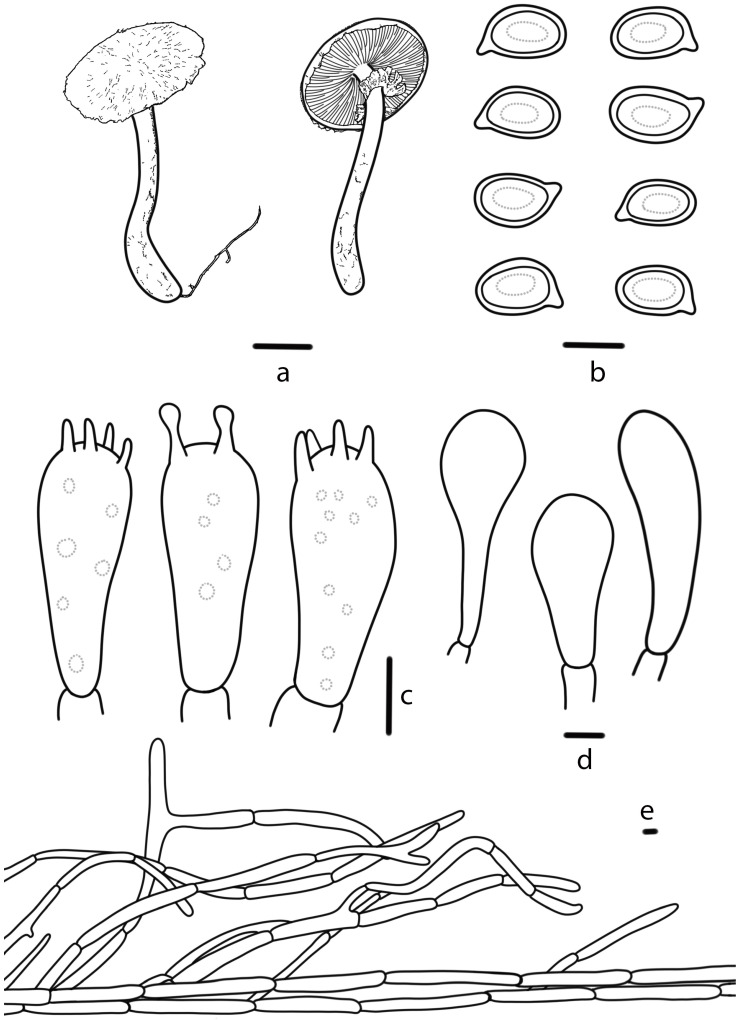
*Agaricus velutinosus* (HMJAU 67764, HMJAU 67768) (**a**) Basidiomata, (**b**) Basidiospores, (**c**) Basidia, (**d**) Cheilocystidia, (**e**) Pileipellis. Bars: (**a**) = 1 cm, (**b**–**e**) = 5 μm.

**Figure 8 jof-10-00059-f008:**
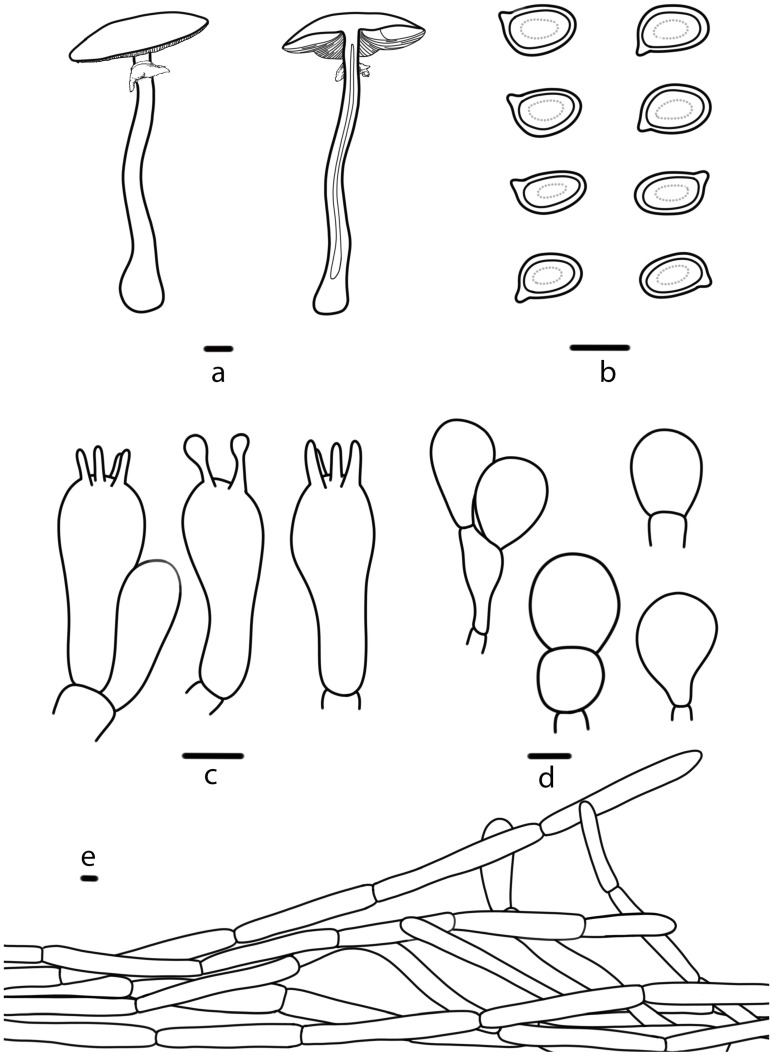
*Agaricus daqinggouensis* (HMJAU 67756, HMJAU 67759) (**a**) Basidiomata, (**b**) Basidiospores, (**c**) Basidia, (**d**) Cheilocystidia, (**e**) Pileipellis. Bars: (**a**) = 1 cm, (**b**–**e**) = 5 μm.

## Data Availability

All the sequences have been deposited in GenBank (https://submit.ncbi.nlm.nih.gov/about/genbank, accessed on 19–20 October 2023) and Mycobank (https://www.mycobank.org, accessed on 15 November 2023); The data presented in this study are deposited in the Fishare (https://figshare.com, accessed on 7 January 2024, accession number, https://doi.org/10.6084/m9.figshare.24631926.v3).

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
