# Peer review of "Six New Species of Agaricus (Agaricaceae, Agaricales) from Northeast China"

_jof, 2024, doi:10.3390/jof10010059_

Round 1
Reviewer 1 Report
Comments and Suggestions for Authors
Many mistakes are repeated along the text which means that the texts have been not rewieved by the authors.
The order of species do not follow alphabetically neither the sections nor the species names.
There are omissions in the comparative taxonomic comments in which differences with some cited taxa are forgotten.
Nomenclature notations are in many cases not fine.
Many times the same character is not uniformly described (in some cases with mistakes) and this denotes that a revision has been not carried out by the authors.
The discusion has nothing to do with the subject of the paper.

The paper needs a major review of the English language, not only in regards to grammar and orthography but also in scientific terminology.
The paper should be reviewed by a mycologist English native speaker.
Author Response
For research article
|
Response to Reviewer 1 Comments
|
||
|
1. Summary |
|
|
|
I would like to express my sincere gratitude for taking the time to review this manuscript. Your expertise and valuable suggestions mean a great deal to me. The feedback you provided in your review was not only detailed but also included clear highlights of the corresponding revisions and corrections made in the resubmitted file. I truly appreciate your thoroughness and patience. Your suggestions and revisions are extremely valuable in enhancing the quality and precision of this manuscript. With your professional guidance, I have acquired a more comprehensive understanding of the issues discussed in the article and have improved my ability to express my thoughts effectively. The necessary modifications have been made and are now highlighted in red font within the text.
|
||
|
2. Questions for General Evaluation |
Reviewer’s Evaluation |
Response and Revisions |
|
Does the introduction provide sufficient background and include all relevant references? |
Yes |
|
|
Are all the cited references relevant to the research? |
Yes |
|
|
Is the research design appropriate? |
Must be improved |
|
|
Are the methods adequately described? |
Can be improved |
|
|
Are the results clearly presented? |
Must be improved |
|
|
Are the conclusions supported by the results? |
Must be improved |
|
|
3. Point-by-point response to Comments and Suggestions for Authors |
||
|
Comments 1: Many mistakes are repeated along the text which means that the texts have been not rewieved by the authors.
|
||
|
Response 1: Thank you for your meticulous editing of the article. I have made the necessary modifications and highlighted them in red font within the text.
|
||
|
Comments 2: The order of species do not follow alphabetically neither the sections nor the species names. |
||
|
Response 2: Thank you for your suggestion; we have revised the article. The species are arranged in alphabetical order, first by sections and then within each section. The most modified in the article are ' Table 1 ', ' Figure 2 ' and ' 3.2 Taxonomy '.
Comments 3: There are omissions in the comparative taxonomic comments in which differences with some cited taxa are forgotten. Response 3: Thank you for your meticulous editing of the article. I have made the necessary modifications and highlighted them in red font within the text.
Comments 4: Nomenclature notations are in many cases not fine. Response 4: Thank you for your suggestion. The genus Agaricus boasts over 500 species, many of which share similar morphological characteristics, posing a challenge in assigning appropriate nomenclature to newly discovered species.
Comments 5: Many times the same character is not uniformly described (in some cases with mistakes) and this denotes that a revision has been not carried out by the authors. Response 5: Thank you for your meticulous editing of the article. I have made the necessary modifications and highlighted them in red font within the text.
Comments 6: The discusion has nothing to do with the subject of the paper. Response 6: Thank you for your suggestion. The discussion in the article was revised by us.
Thank you for your meticulous editing of the article. I have made the necessary modifications and highlighted them in red font within the text. I need to answer some important questions. The relevant information is as follows: Commented [rev.3]: As circumscribed in the paper the genus Agaricus cannot be distinguished from Micropsalliota, Xanthagaricus or Hymenagaricus, all in the family Agaricaceae. Response: Thank you for pointing this out. This content has been modified in the article. ‘Agaricus L. is a large genus of the family Agaricaceae, with Agaricus campestris L. as the type; it could be distinguished from other genera by its unique characteristics, which include small-sized to large-sized fleshy basidiomata, free lamellae which are white or pink when young but at maturity become brown to dark brown, presence of an annulus on the stipe, brown basidiospores, brown spore prints, and pileipellis a cutis of cylindrical hyphae [1–5].’
Commented [rev.4]: The infrageneric ranks are binomials as the species names are and it cannot cited without the genus name. Again it is a widespread mistake, but even if it is widespread remains a mistake. Please look at the Code and you can check that infrageneric names are never abbreviated “section xxxxx” or “sect. xxxxx”. Response: Thank you for pointing this out. The names of all subgenera and sections have been changed to ‘A. subg. XXX’ or ‘A. sect. XXX’.
Commented [rev.5]: I think that it is better to use clades when the authors use branches in the paper. Response: Thank you for pointing this out. Therefore, "the branch" be replaced by "the clade".
Commented [rev.6]: The authors have forgotten A. subg. Conioagaricus Heinem. which is validly published and must not be ruled out until the type can be probed that it is a synonym of a previously published subgenus name. Commented [rev.7]: The authors have overlooked the sections of A. subg. Conioagaricus: A. sect. Intermedii, A. sect. Pulverotecti and A. sect. Striati, and the sections of A. subg, Lanagaricus: A. sect. Lanosi and A. sect. Olicacei. Response: Thank you for pointing this out. ‘In subsequent studies, new subgenera and sections were established based on the new taxonomic system; presently, Agaricus comprises six subgenera and 27 sections [6, 10–14].’ This conclusion is based on new taxonomic system proposed by Zhao et al. (2016). ‘There is a seventh subgenus, A. subg. Conioagaricus, containing three sections: A. sect. intermedii, A. sect. Pulverotecti, and A. sect. Striati [15]. However, due to the members of A. subg. Coniagaricus have an epithelium pileipellis which is completely different from species of the other subgenera, A. subg. Coniagaricus is suggested to be relocated to other genera [2, 16]. Unfortunately, in the absence of available molecular data, A. subg. Conioagaricus has not been studied molecularly within the framework of the new taxonomic system proposed by Zhao et al. (2016) [2]. A. subg. Lanagaricus [15] has been proved a heterotypic synonym of A. subg. Pseudochitonia [2].’ This content has been added in the article.
Commented [rev.10]: In my view the spore length x width average should be also provided in all the new species, as it is in the majority of works in mycology and not only for the Q coeficient Response: Thank you for pointing this out. We have provided the spore length x width average. ‘[Xav = e × f] indicates the average size of basidiospores.’
Commented [rev.19]: Where it is 0.8-1.4 cm thick? Response: Thank you for pointing this out. ‘Context of the pileus up to 0.8–1.4 cm thick, ..’
Commented [rev.21]: This is not a term used in mycology. I guess the authors meant what I have written but if not so they should look for an adequate term to describe the pileus. Response: Agree. Thank you for your careful editing of the article. ‘blunt in center’ means ‘obtusely umbonate at the center’.
Commented [rev.22]: What about griseicephalus also recorded from China. Response: Thank you for pointing this out. We have added this content. ‘A. griseicephalus has a smooth, light brown pileus at maturity, and smaller Qav (Qav = 1.3)’
Commented [rev.23]: Where? Response: Thank you for pointing this out. We have added this content. ‘Agaricus agrocyboides Heinem. & Gooss.-Font. was found in index Fungorum (https://www.speciesfungorum.org/Names/SynSpecies.asp?RecordID=292238).’
Commented [rev.25]: What is this shape? A mirror can have hundreds of shapes. Response: ‘convex mirror-shaped’ means ‘convex-mirror shaped’ or ‘convex’. We chose ‘convex’ instead of ‘convex mirror-shaped’.
Commented [rev.27]: ? Response: Agree. Thank you for your careful editing of the article. ‘long corm-like’ means ‘globose with a long peduncle’.
Commented [rev.28]: ? Response: Agree. Thank you for your careful editing of the article. ‘fan-shaped’ means ‘sphaeropedunculate’.
Commented [rev.31]: Please complete the paragraph as in other species. Response: Agaricus sp. ZRL2012630 was uploaded by Zhao et al. [2], but no formal description was provided. |
||
Commented [rev.34]: These differences with A. arvensis are very weak.
Response: Thank you for pointing this out. ‘Although the macroscopic characteristics of both exhibit minimal differences, discernible disparities exist in their molecular characters.’
The remaining concerns have all been addressed in the article.
|
4. Response to Comments on the Quality of English Language |
|
Point 1: The paper needs a major review of the English language, not only in regards to grammar and orthography but also in scientific terminology. |
|
Response 1: Thank you for your meticulous editing of the article. I have made the necessary modifications and highlighted them in red font within the text. |
|
5. Additional clarifications |
|
Thank you once again for your meticulous editing and thorough review of this article. Best wishes. |

Reviewer 2 Report
Comments and Suggestions for Authors
This manuscript aims to describe six new Agaricus species from Northeast China. It has provided some interesting data. However, their phylogenetic analyses are not sufficient to support their results. Please note that sections such as Xanthodermatei, Minores, and Arvenses are very complicated, and molecular data remain essential for species identification in some cases. Many sequences of the known species are lacking in their dataset. Without them, the phylogenetic positions of those new species remain doubtful.
In more, the authors used the concept of species-specific ITS markers, which requires the dataset containing all available ITS sequences of the section. These alignments should be accessible to readers and reviewers. Could you please provide them?
Figure 2 needs to be improved, in general, we can´t see well the important morphological characters.
It is important to indicate the average size of basidiospores in the description of the new species. In the notes of new species, authors like to differentiate the new species from other known species by using the lamellar color, however, this character does not have too much taxonomic value, especially when you compare sporocarps at different growth stages.
L47, Agaricus subg. Minores contains more than one section; and Agaricus subg. Pseudochitonia consists of 14 sections.
The discussion is very poor. The first four paragraphs are not one discussion. The authors try to say the three new species from A. sect. Arvenses may have potential medicinal and edible values, but this has not been mentioned in other parts of the manuscript. Do local people eat it?
Comments on the Quality of English Language
the English revision is necessary. For example, in the phylogenetic result part (L148), the authors should use four clades, not branches. In many places, "the branch" can be replaced by "the clade". Some expressions are not clear.
Author Response
For research article
|
Response to Reviewer 2 Comments
|
||
|
1. Summary |
|
|
|
I would like to express my sincere gratitude for taking the time to review this manuscript. Your expertise and valuable suggestions mean a great deal to me. The necessary modifications have been made and are now highlighted in red font within the text. Please find the detailed responses below and the corresponding revisions in the resubmitted files.
|
||
|
2. Questions for General Evaluation |
Reviewer’s Evaluation |
Response and Revisions |
|
Does the introduction provide sufficient background and include all relevant references? |
Yes |
|
|
Are all the cited references relevant to the research? |
Yes |
|
|
Is the research design appropriate? |
Can be improved |
|
|
Are the methods adequately described? |
Can be improved |
|
|
Are the results clearly presented? |
Must be improved |
|
|
Are the conclusions supported by the results? |
Must be improved |
|
|
3. Point-by-point response to Comments and Suggestions for Authors |
||
|
Comments 1: This manuscript aims to describe six new Agaricus species from Northeast China. It has provided some interesting data. However, their phylogenetic analyses are not sufficient to support their results. Please note that sections such as Xanthodermatei, Minores, and Arvenses are very complicated, and molecular data remain essential for species identification in some cases. Many sequences of the known species are lacking in their dataset. Without them, the phylogenetic positions of those new species remain doubtful.
|
||
|
Response 1: A. subg. Agaricus has only one section: A. sect. Agaricus. A. sect. Arvenses is the only section of A. subg. Flavoagaricus. A. subg. Minores contains three sections: A. sect. Leucocarpi, A. sect. Minores and A. sect. Pantropicales. A. subg. Pseudochitonia consists of 14 sections. ‘The chromatograms were checked in BioEdit v.7.1.3.0 [35] to ensure that each se-quence had good quality, then a BLAST search was carried in the National Center of Biotechnology Information (NCBI) database (https://www.ncbi.nlm.nih.gov/) to con-firm the sequencing results, and finally the sequences were submitted to GenBank (Table 1 in bold). Based on the BLAST search results, sequences corresponding to the subgenera of the studied species were downloaded for phylogenetic analyses. Subsequently, the multi-gene phylogenetic trees of these subgenera were constructed separately. Particularly, species (Table 1) falling within the clades of the species described in this paper were selected and integrated to construct new multi-gene phylogenetic trees.’ In our phylogenetic analysis, we constructed phylogenetic trees for A. subg. Agaricus (Figure 1.1), A. subg. Flavoagaricus (Figure 1.2), A. subg. Minores (Figure 1.3), and A. subg. Pseudochitonia (Figure 1.4) to ensure the systematic location of these new species. In this study, we constructed the new phylogenetic tree based on previously established phylogenetic trees of each subgenus once these new species were identified. Therefore, these phylogenetic analyses can provide evidence to substantiate the discovery of six novel Agaricus species.
Figure 1.1 Maximum likelihood phylogenetic tree (ML tree) of A. subg. Agaricus based on ITS sequences. Agaricus bisporus LAPAG446 was used as outgroup. Bootstrap support (BS) values ≥50% and Bayesian posterior probability (PP) values ≥0.70 are indicated on clades (BS/PP). 'T' refers to holotype. Bold represents the newly added sequences. The red boxes represent the species selected for this study.
Figure 1.2 Maximum likelihood phylogenetic tree (ML tree) of A. subg. Flavoagaricus based on ITS, nrLSU, and tef1-a sequences. Agaricus edmondoi LAPAG412 was used as outgroup. Bootstrap support (BS) values ≥50% and Bayesian posterior probability (PP) values ≥0.70 are indicated on clades (BS/PP). 'T' refers to holotype. Bold represents the newly added sequences. The red boxes represent the species selected for this study.
|
||
|
Figure 1.3 Maximum likelihood phylogenetic tree (ML tree) of A. subg. Minores based on ITS, nrLSU, and tef1-a sequences. Agaricus campestris LAPAG370 T was used as outgroup. Bootstrap support (BS) values ≥50% is indicated on clades. 'T' refers to holotype. Bold represents the newly added sequences. The red boxes represent the species selected for this study.
Figure 1.4 Maximum likelihood phylogenetic tree (ML tree) of A. subg. Pseudochitonia based on ITS, nrLSU, and tef1-a sequences. Agaricus campestris LAPAG370 T was used as outgroup. Bootstrap support (BS) values ≥50% is indicated on clades. 'T' refers to holotype. Bold represents the newly added sequences. The red boxes represent the species selected for this study. Comments 2: In more, the authors used the concept of species-specific ITS markers, which requires the dataset containing all available ITS sequences of the section. These alignments should be accessible to readers and reviewers. Could you please provide them? |
||
|
Response 2: Thank you for pointing this out. The alignment was submitted to Figshare (https://doi.org/10.6084/m9.figshare.24631926.v2, accessed on 23 December 2023). Those in need can download these alignments from this website. Initially we also uploaded an alignment to the editor.
Comments 3: Figure 2 needs to be improved, in general, we can´t see well the important morphological characters. Response 3: Thank you for pointing this out. Because the journal has certain requirements for the file size of the uploaded manuscripts, we upload clearer photos to the website Figshare (https://doi.org/10.6084/m9.figshare.24631926.v2, accessed on 23 December 2023), which can be downloaded if necessary. Figure 2 has been modified appropriately.
Figure 2. The photographs of fresh basidiomata of Agaricus species in this study.
Comments 4: It is important to indicate the average size of basidiospores in the description of the new species. In the notes of new species, authors like to differentiate the new species from other known species by using the lamellar color, however, this character does not have too much taxonomic value, especially when you compare sporocarps at different growth stages. Response 4: Thank you for pointing this out. We have provided the spore length x width average. ‘[Xav = e × f] indicates the average size of basidiospores.’ The macroscopic and microscopic characteristics of many Agaricus species overlap, therefore, molecular data and phylogenetic analysis are necessary to identify Agaricus species having similar morphological features. Based on this premise, we aim to identify additional morphological distinctions that can differentiate novel species from their closely related species. This character, using the lamellar color, does not have too much taxonomic value. According to suggestions, we have removed this feature in the notes of new species.
Comments 5: L47, Agaricus subg. Minores contains more than one section; and Agaricus subg. Pseudochitonia consists of 14 sections. Response 5: Thank you for pointing this out. A. subg. Minores (Figure 1.3) contains three sections: A. sect. Leucocarpi, A. sect. Minores and A. sect. Pantropicales. A. subg. Pseudochitonia consists of 14 sections (Figure 1.4).
Comments 6: The discussion is very poor. The first four paragraphs are not one discussion. The authors try to say the three new species from A. sect. Arvenses may have potential medicinal and edible values, but this has not been mentioned in other parts of the manuscript. Do local people eat it? Response 6: Thank you for pointing this out. The discussion in the article was revised by us. ‘Agaricus floccularis, A. aurantipileatus and A. griseopileatus are members of A. sect. Arvenses. Currently, A. sect. Arvenses has not been reported to contain any toxic species, with several exhibiting nutritional or medicinal potential, such as cultivated species including A. arvensis Schaeff., A. augustus Fr., and A. subrufescens Peck. [3, 17, 25]. We postulate that A. floccularis, A. aurantipileatus and A. griseopileatus are also edible species that may have potential medicinal and edible values as new cultivatable species. Particularly noteworthy, A. griseopileatus was reported to be edible by some locals, which is precious fungal resources worthy of further excavation and development.’ This content has been modified in the article.
|
||
|
4. Response to Comments on the Quality of English Language |
||
|
Point 1: the English revision is necessary. For example, in the phylogenetic result part (L148), the authors should use four clades, not branches. In many places, "the branch" can be replaced by "the clade". Some expressions are not clear. |
||
|
Response 1: Thank you for pointing this out. I have made the necessary modifications and highlighted them in red font within the text. "the branch" be replaced by "the clade". |
||
|
5. Additional clarifications |
||
|
We have revised the article. For example, the species are arranged in alphabetical order, first by sections and then within each section. The most modified in the article are ' Table 1 ', ' Figure 2 ' and ' 3.2 Taxonomy '. Thank you once again for your suggestions and thorough review of this article. Best wishes. |
||

Round 2
Reviewer 1 Report
Comments and Suggestions for Authors
The authors have made a notable effort to improve the manuscript and errors and deficiencies have generally been adequately corrected making the article desirable for publication.
However, some minor errors are still detected that must be corrected, some of which have occurred during the formatting of the table, leaving some names isolated.
Thus, once these small text and formatting errors have been corrected, I recommend the publication.

Now, the English lanaguage has a much better quality and it is more fluently readable.
Author Response
For research article
|
Response to Reviewer 1 Comments
|
||
|
1. Summary |
|
|
|
I would like to express my sincere gratitude for taking the time to review this manuscript. Your expertise and valuable suggestions mean a lot to me. In your review, not only did you provide detailed feedback, but you also highlighted the corresponding revisions and corrections made in the resubmitted file. I truly appreciate your thoroughness and patience. The suggestions and revisions you have provided are crucial in improving the quality and accuracy of this manuscript. The necessary modifications have been made and are now highlighted in red font within the text.
|
||
|
2. Questions for General Evaluation |
Reviewer’s Evaluation |
Response and Revisions |
|
Does the introduction provide sufficient background and include all relevant references? |
Yes |
|
|
Are all the cited references relevant to the research? |
Yes |
|
|
Is the research design appropriate? |
Yes |
|
|
Are the methods adequately described? |
Yes |
|
|
Are the results clearly presented? |
Yes |
|
|
Are the conclusions supported by the results? |
Yes |
|
|
3. Point-by-point response to Comments and Suggestions for Authors |
||
|
Comments 1: The authors have made a notable effort to improve the manuscript and errors and deficiencies have generally been adequately corrected making the article desirable for publication. However, some minor errors are still detected that must be corrected, some of which have occurred during the formatting of the table, leaving some names isolated. Thus, once these small text and formatting errors have been corrected, I recommend the publication.
|
||
|
Response 1: I am deeply grateful for your assistance. With your professional guidance, this manuscript can be successfully published in the near future. Thank you for your careful editing of the article. I have made the necessary modifications and highlighted them in red font within the text. The revised version is as follows: 1. As here defined Agaricus cannot be distinguished from Micropsalliota. For an unknown reason the authors have not followed my previous correction in which I pointed out that "absence of capitate cheilocystidia" should be added to distinguish Agaricus from Micropsalliota. Response: Thank you for your meticulous editing of the article. "absence of capitate cheilocystidia" has been added.
2. This section is here misplaced out of the table. It shouldbe the first within the table above A. abruptibulbus 3. This section is here misplaced at bottom of the table becasue it is alone and it is the heading for the following species. It should be placed the first in the following page before A. fulvoaurantiacus Response: Thank you for your suggestions. Table 1 has been modified.
4. Agaricus After period the genus names must be written uin full. Response: Thank you for your suggestion. ‘Agaricus’ has been modified.
5. This sentence is better placed at the end of the paragraph becasue the context is described the last in the morphological description. 6. remove here "and" and place it before "context unchanging......" at the end of the paragraph. Response: Thank you for your suggestions. It has been modified.
7. depressed at the center Response: Thank you for your meticulous editing of the article. It has been modified.
8. remove: with fibrillose is an adjective or you write "with fibrils below the annulus or you write fibrillose below the annulus, or you write with fibrillose annulus but not "with fibrillose below the annulus" Response: Thank you for your suggestion. ‘with’ has been removed.
9. Again I must correct this because "curtain-like" is an expression which can be variously interpreted. Better: membranous and pendant Response: Thank you for your meticulous editing of the article. "curtain-like" be replaced by ‘membranous and pendant’.
10. remove this because it is repeated. Response: Thank you for pointing this out. It has been removed.
11. remove here "and" and place it before "context unchanging......" at the end of the paragraph. Response: Thank you for your suggestion. It has been modified.
12. move "when " after fibrils Response: Thank you for your suggestion. It has been modified.
13. remove: 0.6 up to is a limit not a range Response: Thank you for pointing this out. It has been removed.
14. insert with between species and velutinous Response: Thank you for your suggestion. ‘with’ has been added.
15. single, smooth annulus Response: Thank you for your meticulous editing of the article. It has been modified.
16. Remove this sentence which here make not sense Response: Thank you for pointing this out. It has been removed.
17. remove the comma and insert: and Response: Thank you for your suggestions. It has been modified.
|
||
|
4. Response to Comments on the Quality of English Language |
||
|
Point 1: Now, the English language has a much better quality and it is more fluently readable. |
||
|
Response 1: Thank you for your meticulous editing of the article. |
||
|
5. Additional clarifications |
||
|
The manuscript underwent additional modifications, which were indicated in red. Finally, thank you once again for your careful editing and review of this article. Best wishes. |
||
Reviewer 2 Report
Comments and Suggestions for Authors
Dear authors
I am pleased to see significant improvements to this manuscript. Thanks for your efforts. However, I still have numerous observations that need your correction or response. Please find them in the attached pdf.
In addition, as I commented in the first round, the authors used the concept of species-specific ITS markers, which requires the dataset containing all available ITS sequences of the section. The ITS alignment you sent contains only selected sequences of those sections. Thus the positions you reported for your new species can´t represent species-specific markers of the section. You can improve this part by including the missing ITS data (each section to which the new species belongs) or you should remove this part from your manuscript.
About the new species A. aurantipileatus, I have verified your sequence data. HMJAU 67747 and HMJAU 67746 differ at 4 and 5 positions, respectively in their ITS and tef sequences. According to my experience, if they are the true differences, not the heteromorphisms, such a level of genetic divergence is relatively high within the same species. More collections and sequence data are necessary to confirm this genetic variability. Could you please verify the sequence chromatographs of these two collections, and to be sure these differences are true? Could you please provide photos of HMJAU 67746? Thanks

Author Response
For research article
|
Response to Reviewer 2 Comments
|
||||||||||||||||||||||||||||||||||||||||||||||||||||||||||||||||||||||||||
|
1. Summary |
|
|
||||||||||||||||||||||||||||||||||||||||||||||||||||||||||||||||||||||||
|
I would like to express my sincere gratitude for taking the time to review this manuscript. Your expertise and valuable suggestions mean a great deal to me. The feedback you provided in your review was not only detailed but also included clear highlights of the corresponding revisions and corrections made in the resubmitted file. The necessary modifications have been made and are now highlighted in red font within the text.
|
||||||||||||||||||||||||||||||||||||||||||||||||||||||||||||||||||||||||||
|
2. Questions for General Evaluation |
Reviewer’s Evaluation |
Response and Revisions |
||||||||||||||||||||||||||||||||||||||||||||||||||||||||||||||||||||||||
|
Does the introduction provide sufficient background and include all relevant references? |
Yes |
|
||||||||||||||||||||||||||||||||||||||||||||||||||||||||||||||||||||||||
|
Are all the cited references relevant to the research? |
Yes |
|
||||||||||||||||||||||||||||||||||||||||||||||||||||||||||||||||||||||||
|
Is the research design appropriate? |
Yes |
|
||||||||||||||||||||||||||||||||||||||||||||||||||||||||||||||||||||||||
|
Are the methods adequately described? |
Can be improved |
|
||||||||||||||||||||||||||||||||||||||||||||||||||||||||||||||||||||||||
|
Are the results clearly presented? |
Can be improved |
|
||||||||||||||||||||||||||||||||||||||||||||||||||||||||||||||||||||||||
|
Are the conclusions supported by the results? |
Yes |
|
||||||||||||||||||||||||||||||||||||||||||||||||||||||||||||||||||||||||
|
3. Point-by-point response to Comments and Suggestions for Authors |
||||||||||||||||||||||||||||||||||||||||||||||||||||||||||||||||||||||||||
|
Comments 1: In addition, as I commented in the first round, the authors used the concept of species-specific ITS markers, which requires the dataset containing all available ITS sequences of the section. The ITS alignment you sent contains only selected sequences of those sections. Thus the positions you reported for your new species can´t represent species-specific markers of the section. You can improve this part by including the missing ITS data (each section to which the new species belongs) or you should remove this part from your manuscript.
|
||||||||||||||||||||||||||||||||||||||||||||||||||||||||||||||||||||||||||
|
Response 1: Thank you for pointing this out. The ‘2.4 Species-Specific ITS Markers’ experimental method was modified. Relevant modifications are as follows: ‘The positions of ITS markers are sequentially numbered starting from the 5’ end (ggaaggatcatta). The insertion or deletion (indel) in the ITS alignment were disregarded rather than numbered.’ In this manner, whether the ITS alignment contains the dataset of all available ITS sequences in the section, the position of the new species in the species-specific ITS markers remains unaltered. Therefore, the novel positions we have reported for newly discovered species can represent species-specific markers of the section. The ITS alignments of the A. sect. Agaricus, A. sect. Arvenses, A. sect. Minores and A. sect. Xanthodermatei are additionally supplemented. The alignments were submitted to Figshare (https://doi.org/10.6084/m9.figshare.24631926.v3, accessed on 7 January 2024). Those in need can download these alignments from this website.
|
||||||||||||||||||||||||||||||||||||||||||||||||||||||||||||||||||||||||||
|
Comments 2: About the new species A. aurantipileatus, I have verified your sequence data. HMJAU 67747 and HMJAU 67746 differ at 4 and 5 positions, respectively in their ITS and tef sequences. According to my experience, if they are the true differences, not the heteromorphisms, such a level of genetic divergence is relatively high within the same species. More collections and sequence data are necessary to confirm this genetic variability. Could you please verify the sequence chromatographs of these two collections, and to be sure these differences are true? Could you please provide photos of HMJAU 67746? Thanks |
||||||||||||||||||||||||||||||||||||||||||||||||||||||||||||||||||||||||||
|
Response 2: Thank you for pointing this out. We reverify the sequence chromatographs of these two collections. The results showed that the quality of the sequence chromatographs (ITS and tef1-a) of the HMJAU 67747 collection was not as good as HMJAU 67746 collection. This may have resulted in an error in sequencing. The HMJAU 67747 collection and other collections are also distinguished in Table 1 and Table 2. In the polygenic tree (Figure 1), A. aurantipileatus, A. subumbonatus R.L. Zhao & B. Cao and A. sp. ZRL2012630 formed a larger clade with a high support value. The three are related species. We analyzed the situation when HMJAU 67746 and HMJAU 67747 sequences were different in the ITS and tef1-a alignments. The ITS alignment starting from the 5’ end (ggaaggatcatta) (Table 1). The tef1-a alignment starting from the 5’ end (aggctgactgtgc) (Table 2). The insertion or deletion (indel) in the alignments were disregarded rather than numbered. The nrLSU sequences of HMJAU 67746 and HMJAU 67747 collections are the same. There were four differences in the ITS alignment between the HMJAU 67747 and HMJAU 67746 collections (Table 1). The validity of three of these differences (117, 249, and 351) may be questionable due to the sequence chromatograph of the HMJAU 67747 collection was not very good and showed specificity. There were six differences in the tef1-a alignment between the HMJAU 67747 and HMJAU 67746 collections (Table 2). Considering the conservative nature of the initial tef1-a alignment, ' C ' of HMJAU 67746 sequence at the 2 site should be replaced with ' G '. The ' Y ' of HMJAU 67747 sequence at the 128 site is a degenerate base (Y = C/T). The validity of three of these differences (365 and 369) may be questionable due to the sequence chromatograph of the HMJAU 67747 collection was not very good and showed specificity. In summary, there were one (Table 1, 178) actual differences in the ITS alignment and two (Table 2, 308 and 320) differences in tef1-a alignment between HMJAU 67746 and HMJAU 67747 collections. And we did not find any obvious morphological differences between them. Therefore, both are here considered conspecific.
We have provided photos of HMJAU 67746 (Figure 1.1).
Table 1. Genotypes at 5 loci of ITS in 2 collections of A. aurantipileatus, 1 collection of A. sp. ZRL2012630 and 2 collections of A. subumbonatus.
Table 2. Genotypes at 6 loci of tef1-a in 2 collections of A. aurantipileatus, 1 collection of A. sp. ZRL2012630 and 2 collections of A. subumbonatus.
Figure 1.1 The photographs of Agaricus aurantipileatus HMJAU 67746. Bars: (a–b) =1 cm.
Thank you for your meticulous editing of the article. We have made the necessary modifications and highlighted them in red font within the text. We need to answer some important questions. The relevant information is as follows: 1. please add explanations for e and f Response: Thank you for your suggestion. It is a pity that we don't understand what 'e' and 'f' refer to, so there is no modification. If necessary, we will further modify.
2. for the genus Agaricus, chemical tests as KOH and Schaeffer’s reaction are important. Did you make it? Response: Agree. It is a pity that we were unable to do it, and we hope to make it up in the future, or conduct tests on dry specimens. Additionally, as reported by Zhao et al. (2016) [2], the outcomes of KOH and Schaeffer's reaction for species within the same section exhibit congruence. For example, A. sect. Agaricus: KOH and Schäffer’s reactions negative; A. sect. Minores: KOH and Schäffer’s reactions positive; A. sect. Xanthodermatei: KOH reaction positive yellow, Schäffer's reaction negative. The new species should possess the distinctive characteristics inherent to their respective sections. At present, these phylogenetic analyses and morphological characteristics can provide evidence to substantiate the identify of six novel Agaricus species.
3. Could you please rewrite this sentence, which is very long and unclear. Response: Thank you for your suggestion. It has been modified.
4. Did you set this value from the beginning? Response: We initially set ‘1, 000, 000 generations’, but terminated at ‘743,400 generations’ due to the split deviation frequency value (0.009756) was <0.01.
5. better change to "type specimen" because in the case of LAPAG450, it is the epitype rather than the holotype. Response: Thank you for your suggestion. It has been changed to ‘type specimen’.
6. you mean gaps? Response: Thank you for pointing this out. It has been modified. (See Response 1.)
7. the same or very similar? Response: Thank you for pointing this out. It is ‘very similar’.
8. please add "as follows:" The complete sentence will be "Six new species are distributed in these four sections as follows:" 9. please insert "in A. sect. Xanthidermatei" after (BS/PP = 84/0.80) 10. please insert in A. sect. Agaricus" 11. please add "in A. sect. Minores" 12. please add "sequence data" Response: Thank you for your meticulous editing of the article. They have been added separately.
13. when young? How about when it mature? Response: Thank you for your suggestion. It has been modified. ‘pale red (10A3) when young, reddish brown (8E8) or slightly purplish pink (14A3) at mature,’
14. do you mean above the annulus? Response: Yes. It has been modified. ‘pale red (10A3) above the annulus sometimes,’
15. sequence of ZRL2012630 Response: Thank you for your meticulous editing of the article. It have been added.
16. what are the differences? how many bps? Response: Thank you for pointing this out. It has been modified. (See Response 2.)
17. all these are when bruised? Response: It has been modified. ‘white (4A1), yellowish white (4A2), straw yellow (3B4), hollow, dark yellow (4C8) on touching or bruising,’
18. please check the whole manuscript, use only odor or odour Response: Thank you for pointing this out. The ‘odour’ be replaced by ‘odor’ in the whole manuscript.
19. any discoloration on handing or cutting?
Figure 2.1 The photograph of A. velutinosus HMJAU 67769.
Figure 2.2 The photograph of A. velutinosus HMJAU 67767.
20. please add "for example, A. sect. Arvenses and A. sect. Xanthodermatei" 21. please add "some sections of" Response: Thank you for your meticulous editing of the article. They have been added separately.
22. it is too general to avoid all wild Agaricus. It could be better to say "unknown wild Agaricus" Response: Thank you for your suggestion. It has been modified. ’unknown wild Agaricus species’
|
||||||||||||||||||||||||||||||||||||||||||||||||||||||||||||||||||||||||||
|
4. Response to Comments on the Quality of English Language |
||||||||||||||||||||||||||||||||||||||||||||||||||||||||||||||||||||||||||
|
Point 1: |
||||||||||||||||||||||||||||||||||||||||||||||||||||||||||||||||||||||||||
|
Response 1: |
||||||||||||||||||||||||||||||||||||||||||||||||||||||||||||||||||||||||||
|
5. Additional clarifications |
||||||||||||||||||||||||||||||||||||||||||||||||||||||||||||||||||||||||||
|
The manuscript underwent additional modifications, which were indicated in red. For example, ' Table 1 '. Finally, thank you once again for your careful editing and review of this article. Best wishes. |
||||||||||||||||||||||||||||||||||||||||||||||||||||||||||||||||||||||||||
